# Mitochondrial injury induced by a *Salmonella* genotoxin triggers the proinflammatory senescence-associated secretory phenotype

Han-Yi Chen[1], Wan-Chen Hsieh [2], Yu-Chieh Liu[1], Huei-Ying Li[3], Po-Yo Liu[1], Yu-Ting Hsu[1], Shao-Chun Hsu [4], An-Chi Luo[4], Wei-Chen Kuo[5], Yi-Jhen Huang[6], Gan-Guang Liou[7], Meng-Yun Lin[8], Chun-Jung Ko [8], Hsing-Chen Tsai [6,9,10] & Shu-Jung Chang [1] ✉

Bacterial genotoxins damage host cells by targeting their chromosomal DNA. In the present study, we demonstrate that a genotoxin of *Salmonella* Typhi, typhoid toxin, triggers the senescence-associated secretory phenotype (SASP) by damaging mitochondrial DNA. The actions of typhoid toxin disrupt mitochondrial DNA integrity, leading to mitochondrial dysfunction and disturbance of redox homeostasis. Consequently, it facilitates the release of damaged mitochondrial DNA into the cytosol, activating type I interferon via the cGAS-STING pathway. We also reveal that the GCN2-mediated integrated stress response plays a role in the upregulation of inflammatory components depending on the STING signaling axis. These SASP factors can propagate the senescence effect on T cells, leading to senescence in these cells. These findings provide insights into how a bacterial genotoxin targets mitochondria to trigger a proinflammatory SASP, highlighting a potential therapeutic target for an anti-toxin intervention.

Typhoid toxin is a genotoxin first discovered in *Salmonella enterica* serovar Typhi (*S.* Typhi) and Paratyphi A[1], which cause systemic illnesses in humans known as typhoid fever and paratyphoid fever[2,3]. This class of bacterial genotoxins serves as pivotal virulence factors, triggering DNA damage, cell cycle arrest, and eventually cell death, thereby substantially contributing to disease development[4,5]. In a murine model, the administration of typhoid toxin induces mortality and recapitulates many characteristic symptoms of typhoid fever[1].

However, ethical considerations limit its in-depth investigation in human volunteer studies, primarily because these studies tend to focus predominantly on the acute phase of typhoid fever[6–8]. Recently, whole genome-sequence studies have revealed that over 40 clade B nontyphoidal *Salmonella* (NTS) serovars produce typhoid toxin, including a highly virulent serotype Javiana (*S.* Javiana)[9]. In mice, typhoid toxin promotes systemic infection of *S.* Javiana, highlighting its crucial role in the context of bacterial infection[9].

[1]Graduate Institute of Microbiology, College of Medicine, National Taiwan University, Taipei, Taiwan. [2]Institute of Molecular and Cellular Biology, National Tsing Hua University, Hsinchu, Taiwan. [3]Medical Microbiota Center of the First Core Laboratory, College of Medicine, National Taiwan University, Taipei, Taiwan. [4]Imaging Core, College of Medicine, National Taiwan University, Taipei, Taiwan. [5]Institute of Biochemistry and Molecular Biology, College of Medicine, National Taiwan University, Taipei, Taiwan. [6]Graduate Institute of Toxicology, College of Medicine, National Taiwan University, Taipei, Taiwan. [7]Cryo-EM Core, College of Medicine, National Taiwan University, Taipei, Taiwan. [8]Graduate Institute of Immunology, College of Medicine, National Taiwan University, Taipei, Taiwan. [9]Department of Internal Medicine, National Taiwan University Hospital, Taipei, Taiwan. [10]Center for Frontier Medicine, National Taiwan University Hospital, Taipei, Taiwan. ✉e-mail: sjchang@ntu.edu.tw

These findings underscore the significance of typhoid toxin in bacterial pathogenesis.

Typhoid toxin has a unique biology, as it is exclusively produced by intracellular *Salmonella* and delivered to the extracellular space via a receptor-mediated exocytic transport pathway[10–12]. Through autocrine and paracrine routes, typhoid toxin enters various cell types via receptor-mediated retrograde transport to its subcellular destination[13]. Typhoid toxin comprises two enzymatic subunits, CdtB and PltA[1]. The latter is linked to a homopentameric complex composed of the binding subunit, PltB or PltC, which mediates toxin internalization into host cells[1,14]. CdtB, with its nuclease activity, inflicts DNA damage that results in the arrest of cell proliferation in intoxicated cells. DNA damage instigates host DNA repair processes to maintain genome integrity[10]. Emerging evidence has shown that typhoid toxin activates a non-canonical DNA damage response termed RING (response induced by a genotoxin), which requires single-stranded DNA sensing via its binding protein complex RPA[15]. The activation of RING induced by typhoid toxin triggers cellular senescence by detecting the senescence-associated β-galactosidase (SA-β-gal) activity[15]. When cells become senescent, they may generate the SASP, a phenomenon characterized by the production and secretion of various soluble factors such as proinflammatory cytokines, chemokines, proteases, and metalloproteinases into the extracellular milieu, which can modulate intercellular communication[16]. However, the unrestrained SASP and paracrine senescence in neighboring cells can be detrimental to tissue homeostasis. This is because a substantial rise in proinflammatory modulators can hinder the process of wound healing and tissue repair[17]. Interestingly, patients suffering from typhoid fever often experience rigorous inflammatory responses and tissue damage, leading to severe complications such as leukopenia, lymphopenia, and intestinal perforation[7]. These complications can arise as a consequence of the interplay between pathogen virulence factors and the host's responses to *S*. Typhi infection.

In this study, we investigated the mechanism underlying the typhoid toxin-induced SASP. Our findings uncover that typhoid toxin triggers the SASP through mitochondrial genome impairment and the associated mitochondrial dysfunction. Subsequently, the release of mtDNA into the cytosol activates the cytosolic DNA-sensing STING signaling pathway, resulting in the upregulation of type I interferon (IFN) and the induction of the GCN2-mediated integrated stress response (ISR). The latter process contributes to the expression of proinflammatory SASP components. Ultimately, our discovery suggests a link between typhoid toxin-induced SASP and adaptive immunity, as SASP factors secreted by senescent macrophages induce a bystander effect in T cells. These immune effects could impede host defense and sustain chronic inflammation during *S*. Typhi infection.

## Results

### Typhoid toxin induces cellular senescence and proinflammatory SASP in macrophages and intestinal epithelial cells

Immune cells, including macrophages, play a critical role in shaping immune responses in reaction to pathogenic invasions and tissue damage. Recent research also revealed that typhoid toxin produced by *S*. Typhi can promote bacterial infection in epithelial cells via the induction of cellular senescence[15,18]. To explore whether typhoid toxin induces senescence in macrophages, we investigated the activation and expression of senescence markers in THP-1-derived macrophages exposed to wild-type (WT) typhoid toxin and compared them to cells treated with a mutant toxin variant (PltB[S35A]), which cannot bind to cell surface receptors due to a single amino acid substitution in the glycan-binding domain of the PltB subunit. Our observations revealed the activation of the DNA damage marker γ-H2AX, phosphorylation of p53, and the induction of p16[INK4a],

well-recognized senescence markers (Fig. 1a). Importantly, the senescence markers p53 and p16[INK4a] became significantly evident only after 16 h of typhoid toxin treatment, coinciding with a substantial increase in mRNA levels encoding proinflammatory components (Supplementary Fig. 1a). Furthermore, we also detected an elevation in the expression of selected proinflammatory SASP components linked to the SASP (Supplementary Fig. 1b). This observation points to the onset of cellular senescence and its timing of induction. The induction of senescence was further substantiated by heightened SA-β-gal activity in THP-1-derived macrophages exposed to WT typhoid toxin (Fig. 1b, c). Given the timing of senescence activation, we conducted transcriptome analysis of THP-1-derived macrophages treated with WT typhoid toxin 16 h post-treatment, comparing it to the control group (the PltB[S35A] mutant). RNA-seq analysis revealed differential expression of 503 genes, with a minimum 1.5-fold change (298 upregulated and 205 down-regulated), all with a *P*-value < 0.05 cutoff in WT typhoid toxin-treated cells (Fig. 1d, e). Consistent with the earlier observations regarding typhoid toxin, gene set enrichment analysis (GSEA) uncovered a reduction in the cell cycle pathway in WT typhoid toxin-treated cells, which is linked to cellular senescence. Furthermore, GSEA demonstrated a noteworthy enrichment in pathways linked to cellular senescence and its associated activities[19], encompassing adherens junctions, focal adhesions, the peroxisome proliferator-activated receptors (PPAR) signaling pathway, and the nucleotide-binding oligomerization domain (NOD)-like receptor signaling pathway (Fig. 1f, g). GSEA also highlighted a significant enrichment of the chemokine signaling pathway and cytokine activity in cells treated with WT typhoid toxin (Fig. 1g and Supplementary Fig. 1c, d). In line with Freund et al.'s proposed SASP factors, we also identified a subset of SASP factors significantly upregulated in cells exposed to WT typhoid toxin (Fig. 1h). We assessed the secretion of inflammatory components using a cytokine array (Fig. 1i) and ELISA (Supplementary Fig. 1e), aligning with the phenotypic characteristics associated with a proinflammatory SASP response.

We noticed that the expression of IL-8, IL-1β, and TNF-α became detected as early as 8 h in our time course analyses. Considering the timing of γ-H2AX, phosphorylation of p53 and induction of p16[INK4a], it is reasonable to infer that at least two senescence-related signaling pathways contribute to the production of proinflammatory components in typhoid toxin-treated cells. Interestingly, we observed that inhibiting the ataxia-telangiectasia mutated (ATM) protein, responsible for phosphorylating H2AX (γ-H2AX) in response to DNA damage, partially reduced the expression of IL-8, IL-1β, and TNF-α following typhoid toxin treatment at 8 h, but not 16 h (Supplementary Fig. 2a, b). Moreover, cells lacking ATM still exhibited positive SA-β-gal activity (Supplementary Fig. 2c–e). These results suggest that typhoid toxin induces an unidentified pathway that triggers the SASP phenotype in cells. To validate the presence of SASP in primary cells, we verified SASP characteristics by detecting elevated mRNA levels of *Cxcl2* (murine IL8 homolog), *Il1b*, *Il18*, *Tnf*, and *Il6* in primary murine bone marrow-derived M0 macrophages (BMDMs) after exposure to WT typhoid toxin (Fig. 1j). To explore the influence of M1 and M2 macrophage subtypes on the production of these SASP components, we assessed the mRNA levels of these selected SASP factors in BMDMs stimulated with IFN-γ and LPS or IL-4 and IL-13, which drive M1 and M2 activation, respectively. Our findings indicated that both M1 and M2 macrophages produced CXCL2, IL-6, and TNF-α following treatment with WT typhoid toxin (Supplementary Fig. 3). It is worth noting that M1 macrophages exhibited a more pronounced upregulation of IL-1β, while M2 macrophages promoted the expression of IL-18 (Supplementary Fig. 3). These results suggest distinctions in the extent to which different macrophage subtypes contribute to the production of inflammatory SASP components. We further investigated the generalizability of the observed typhoid

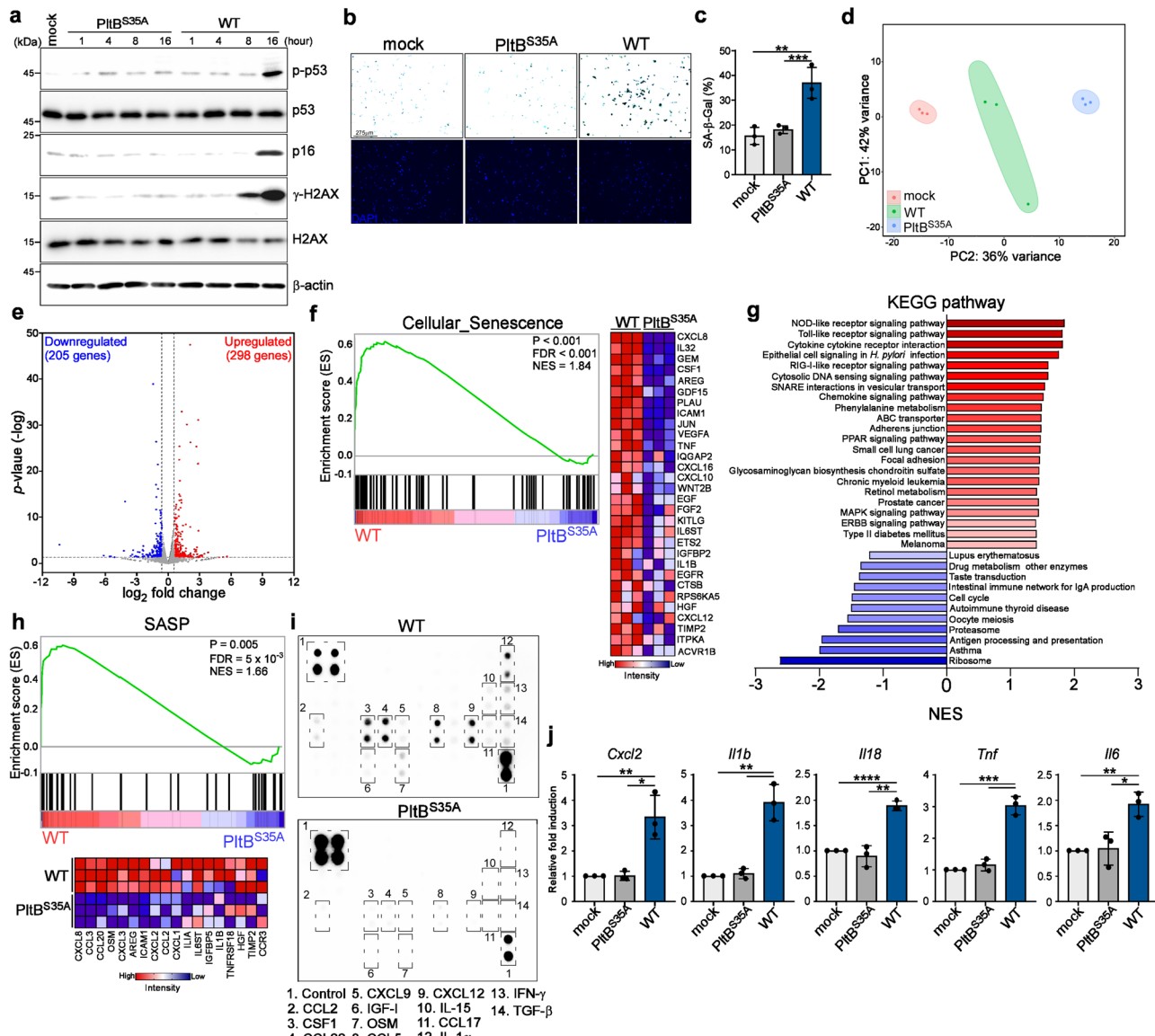

**Fig. 1 | Typhoid toxin induces cellular senescence and proinflammatory SASP in cells. a** Western blot analysis of THP-1-derived macrophages exposed to typhoid toxin. THP-1-derived macrophages were treated with either wild-type (WT) typhoid toxin or the PltB$^{S35A}$ mutant at 37 °C for 1 h and then changed to regular growth medium. The cell lysates were collected at indicated time points and subjected to western blot analysis for protein assessment. **b, c** THP-1-derived macrophages exposed to typhoid toxin exhibited SA-β-gal activity staining. DAPI staining is shown in blue. Scale bars, 0.275 mm. The quantification results are presented as mean ± s.d ($n$ = 3) (**c**). Statistical analysis was performed using unpaired two-sided $t$-tests; **$P$ < 0.01, ***$P$ < 0.001. **d** Principal component analysis (PCA) of transcriptome data for THP-1-derived macrophages exposed to typhoid toxin. **e** A volcano plot showing gene expression changes in WT typhoid toxin-treated macrophages compared to PltB mutant toxin-treated macrophages, highlighting upregulated genes in red and down-regulated genes in blue ($P$ < 0.05 and fold change >1.5). **f** Gene set enrichment analysis (GSEA) showing upregulated genes related to cellular senescence in typhoid toxin-treated macrophages compared

with the PltB$^{S35A}$ mutant. A heatmap represents the gene expression levels involved in cellular senescence as determined by RNA-seq. **g** Additional GSEA was conducted to show Kyoto Encyclopedia of Genes and Genomes (KEGG) pathway enrichment in WT typhoid toxin-treated macrophages versus the PltB$^{S35A}$ mutant, with normalized enrichment scores (NES). **h** GSEA showing upregulated genes related to the senescence-associated secretory phenotype (SASP) in WT typhoid toxin-treated macrophages compared to the PltB$^{S35A}$ mutant. A heatmap represents gene expression levels involved in the SASP as determined by RNA-seq. **i** Cytokine profile of conditional medium obtained from THP-1-derived macrophages exposed to WT typhoid toxin or the PltB$^{S35A}$ mutant. The SASP factors are enclosed with labels. **j** RT-qPCR analysis of mRNA levels of indicated genes in murine bone marrow-derived macrophages (BMDMs) exposed to WT typhoid toxin and the PltB$^{S35A}$ mutant. Data are presented as mean ± s.d ($n$ = 3). Statistical analysis was performed using unpaired two-sided $t$-tests; *$P$ < 0.05, **$P$ < 0.01, ***$P$ < 0.001, ****$P$ < 0.0001. The western blots shown in (**a**) are representative of 3 independent experiments. Source data are provided as a Source data file.

toxin-induced SASP by conducting experiments on Henle-407 intestinal epithelial cells. Our findings indicated positive staining for SA-β-gal activity, elevated mRNA expression levels of SASP factors, and increased p16$^{INK4a}$ and p53 activation (Supplementary Fig. 4). Taken together, these results provide strong evidence that typhoid toxin has the capacity to induce cellular senescence and the development of the SASP across different cell types.

## The cGAS-STING signaling pathway plays a central role in the proinflammatory SASP induced by typhoid toxin

To explore potential cellular pathways implicated in the induction of proinflammatory SASP by typhoid toxin, we analyzed RNA-seq data with a particular focus on immunoregulatory signaling. Our findings revealed a significant enrichment of genes associated with cytosolic DNA sensing in cells treated with WT typhoid toxin, as compared to

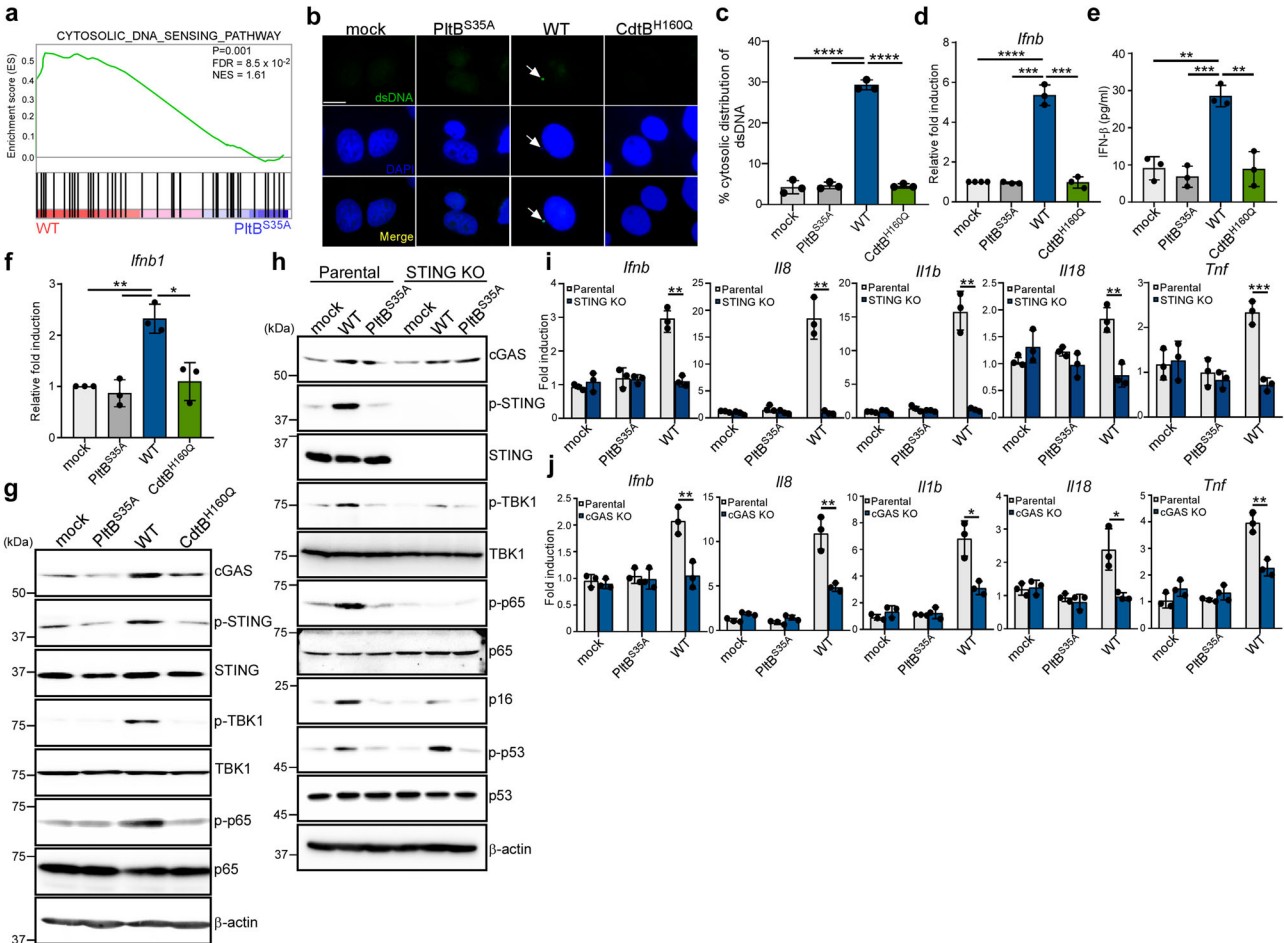

**Fig. 2 | The cGAS-STING signaling pathway plays a central role in the proin-flammatory SASP induced by typhoid toxin.** **a** GSEA showing upregulated genes related to cytosolic DNA sensing pathway in macrophages treated with wild-type (WT) typhoid toxin compared to those treated with the PltB[S35A] mutant. **b** Representative images of immunofluorescent staining in Henle-407 cells treated with WT typhoid toxin, the PltB[S35A] and the CdtB[H160Q] mutants with anti-dsDNA and DAPI are shown in green and blue, respectively. Arrowheads indicate the presence of cytosolic dsDNA. Scale bars, 12.5 μm. **c** Percentage of cytoplasmic distribution of dsDNA 24 h after treatment of typhoid toxin in Henle-407 cells. Values are shown as mean ± s.d. (*n* = 3). One hundred cells were counted for each condition. Statistical analysis was performed using unpaired two-sided *t*-tests; ****$P < 0.0001$. **d** RT-qPCR analysis in THP-1-derived macrophages exposed to typhoid toxin. Statistical analysis was performed using unpaired two-sided *t*-tests (*n* = 3); ***$P < 0.001$, ****$P < 0.0001$. **e** IFN-β protein levels were assessed using ELISA in THP-1-derived macrophages after exposure to WT typhoid toxin and the mutants for 16 h. Data are presented as mean ± s.d (*n* = 3). Statistical analysis was performed using unpaired

two-sided *t*-tests; **$P < 0.01$, ***$P < 0.001$. **f** RT-qPCR analysis in murine bone marrow-derived macrophages exposed to typhoid toxin. Statistical analysis was performed using unpaired two-sided *t*-tests (*n* = 3); *$P < 0.05$, **$P < 0.01$. **g** THP-1-derived macrophages exposed to typhoid toxin were analyzed using western blot analysis with antibodies against cGAS, phosphorylated STING, phosphorylated TBK1, phosphorylated p65, and β-actin as a loading control. **h** Analysis of the STING signaling pathway was performed using the STING-deficient THP-1-derived mac-rophages and its parental cells exposed to typhoid toxin. Cell lysates were analyzed by western blot with antibodies against cGAS, phosphorylated STING, phos-phorylated TBK1, phosphorylated p65, p16[INK4a], phosphorylated p53, and β-actin as a loading control. **i, j** The mRNA levels of proinflammatory components were assessed using RT-qPCR in the parental, STING knockout (KO) (**i**) or cGAS KO (**j**) macrophages exposed to typhoid toxin. Statistical analysis was performed using unpaired two-sided *t*-tests (*n* = 3); *$P < 0.05$, **$P < 0.01$ ***$P < 0.001$. The western blots shown in (**g**), (**h**) are representative of 3 independent experiments. Source data are provided as a Source data file.

those treated with the PltB[S35A] mutant (Fig. 2a). Immunofluorescence staining confirmed the accumulation of cytosolic DNA in cells exposed to WT typhoid toxin (Fig. 2b, c), which was absent in cells exposed to a DNase-I inactive mutant version of the toxin (CdtB[H160Q])[1,10]. This finding compellingly suggests that CdtB-mediated DNA damage results in the release of DNA leakage into the cytosol, where cytosolic DNA can function as a danger-associated molecular pattern (DAMP) and acti-vate proinflammatory SASP through the DNA sensing cGAS-STING axis[20,21]. Moreover, we found that WT typhoid toxin treatment led to an increased mRNA level of *Ifnb* and the secretion of IFN-β in THP-1-derived macrophages and primary BMDMs (Fig. 2d–f), whereas the PltB[S35A] or CdtB[H150Q] mutants did not elicit the same response (Fig. 2d–f, Supplementary Fig. 5). We also noted that WT typhoid toxin activated the expression of the IFN-β promoter-driven reporter in intestinal epithelial cells (Supplementary Fig. 6). Coincidently, we observed a

substantial increase in cGAS protein expression, along with the phos-phorylation of STING and TANK-binding kinase-1 (TBK1) in cells treated with WT typhoid toxin (Fig. 2g). This cascade can activate the NF-κB p65 signaling pathway (Fig. 2g), a well-recognized primary driver of SASP for stimulating the expression of proinflammatory genes encoding cytokines and chemokines[22]. Inactivation of STING led to the suppression of the NF-κB pathway as well as mRNA levels encoding IFN-β and other proinflammatory components (Fig. 2h–j, Supplemen-tary Fig. 7). A similar effect was also observed in cGAS-deficient cells. While SA-β-gal activity remained detectable in cGAS-deficient cells following typhoid toxin treatment (Supplementary Fig. 8), compro-mised SA-β-gal activity and reduced expression of p16[INK4a] was evi-dent in cells lacking STING (Fig. 2h). Of note, STING deficiency did not prevent the phosphorylation of p53 (Fig. 2h). These findings suggest that typhoid toxin activates the p53 and p16[INK4a]-mediated senescence

pathways, with STING primarily influencing the p16^INK4a-mediated senescence pathway and the SASP in response to typhoid toxin.

Our RNA-seq data also revealed an upregulation of genes associated with the retinoic acid-inducible gene I (RIG-I)-like receptor RNA-sensing signaling pathway in WT toxin-treated cells (Supplementary Fig. 9a). Nevertheless, the inactivation of mitochondrial antiviral signaling protein (MAVS), a downstream adaptor molecule of RIG-I that mediates the expression of IFN-β via the IRF3 and NF-κB pathways[23], did not abolish the expression of IFN-β and other proinflammatory components (Supplementary Fig. 9b, c). This observation indicates that RIG-I-MAVS signaling is not involved in activating proinflammatory SASP induced by typhoid toxin. Collectively, our results emphasize the centrality of the cGAS-STING-TBK1 axis as the key regulator in the induction of SASP by typhoid toxin.

## Typhoid toxin activates the cGAS-STING signaling pathway by promoting mtDNA efflux into the cytosol

The cGAS-STING signaling pathway detects cytoplasmic DNA that may originate from damaged nuclear or mitochondrial DNA[20]. Interestingly, electron microscopy images revealed that cells exposed to typhoid toxin had a large number of mitochondria with loss of cristae or cristae disorganization and swollen morphology (Fig. 3a). In addition, in cells treated with WT typhoid toxin, the mitochondria became fragmented, while those treated with either the PltB or CdtB mutant did not exhibit this effect (Fig. 3b, c). These findings collectively point to typhoid toxin-induced mitochondrial dysfunction and an imbalance in osmotic homeostasis. By using the cell-permeant dye JC-1, we detected a loss of mitochondrial membrane potential only at the 16-h mark after toxin treatment. The loss was evidenced by an increase in green fluorescent monomers and a decrease in JC1 aggregates (Fig. 3d, e, and Supplementary Fig. 10), providing further evidence of typhoid toxin-induced mitochondrial dysfunction. In light of these observations, we reasoned that typhoid toxin may directly impair mitochondria, thus prompting the release of mtDNA into the cytosol. Consistent with our hypothesis, we noted that WT typhoid toxin led to a noticeable increase in the intensity of cytosolic DNA, which was associated with transcriptional factor A (TFAM), a main component of mtDNA nucleoids (Fig. 3f, g). Further supporting this notion, we observed elevated levels of cytosolic mtDNA (cmtDNA) in cells exposed to WT typhoid toxin, while cells treated with the CdtB mutant exhibited cmtDNA levels similar to those of mock or PltB mutant-treated cells (Fig. 3h, i). These findings provide strong evidence of a significant accumulation of cytosolic DNA originating from the mitochondria.

To further investigate the direct involvement of mtDNA in typhoid toxin-induced proinflammatory SASP, we used a nucleoside analog 2',3'-dideoxycytidine (ddC) to deplete mtDNA in cells exposed to typhoid toxin (Fig. 3j). Remarkably, depleting mtDNA significantly reduced levels of mRNA encoding IFN-β and proinflammatory components (Fig. 3k), providing compelling evidence that SASP induced by typhoid toxin occurs through the mtDNA-dependent immune response. Therefore, these findings establish a connection between the release of mtDNA from damaged mitochondria and the emergence of typhoid toxin-induced proinflammatory SASP.

## Typhoid toxin interaction with mitochondrial DNA and its induction of damage

Considering the role of CdtB in the DNA damage response within the nuclear genome, we postulated that typhoid toxin directly affects mtDNA, resulting in mtDNA damage and subsequent degradation. This hypothesis was supported by the localization of a substantial fraction of the CdtB subunit of typhoid toxin in the purified mitochondria (Fig. 4a). Furthermore, the protease susceptibility experiment indicated that the localization of CdtB within internal mitochondria (Supplementary Fig. 11a), further substantiating our

hypothesis that typhoid toxin enters the mitochondria to cause mitochondrial damage. To explore whether typhoid toxin indeed targets mtDNA directly, we conducted a mitochondrial genome-wide ChIP assay (mtDIP) to assess interactions between typhoid toxin and mtDNA. Primer pairs covering the entire mitochondrial genome were used to amplify segments approximately every 300–400 bp of mtDNA, with TFAM serving as a positive control. Remarkably, RT-qPCR analyses revealed a noteworthy increase in CdtB binding to the mitochondrial displacement loop (D-loop) and regions spanning nucleotide positions between 4735 and 5172, as well as 8459 and 11,802 in cells exposed to WT typhoid toxin compared to those treated with a mock or the PltB^S35A mutant (Fig. 4b). Importantly, CdtB co-occupied the D-loop region with TFAM at comparable or even higher levels within the mitochondria (Fig. 4b). As expected, there was a significant increase in mtDNA damage and a noticeable reduction in mtDNA copy number in cells treated with the WT toxin (Fig. 4c, d). In contrast, mock and the PltB mutant-treated cells did not exhibit these effects (Fig. 4c, d). These results confirm the occurrence of mtDNA lesions and robustly support the assertion that mtDNA is a direct target of typhoid toxin.

Among bacterial genotoxins, cytolethal distending toxin (CDT) is present in various pathogenic bacteria, such as *C. jejuni*. CDT is composed of three subunits: CdtA and CdtC, forming its heterodimeric B subunit, and CdtB, functioning as the single A subunit, which shares homology with the CdtB subunit of typhoid toxin. Previous research has confirmed that CDT induces senescence and the SASP through the ATM-p38 signaling axis[24]. However, whether CDT provokes damage to mtDNA remains uncertain. In our observations with macrophages, we noted that CDT led to increased mRNA levels of proinflammatory components (Fig. 4e), although it did not include IFN-β. In addition, there was no discernible decrease in mtDNA copy number, nor was cytosolic mtDNA detected in cells exposed to CDT (Fig. 4f, g). Moreover, CDT was unable to access the mitochondria (Supplementary Fig. 11b). These findings indicate that CDT activates the SASP through a mechanism distinct from that of typhoid toxin.

## Typhoid toxin-induced mitochondrial injury leads to the production of mitochondrial ROS (mtROS) to trigger the cytosolic release of mtDNA

We next delved into the mechanism underlying the release of mtDNA from stressed mitochondria during typhoid toxin intoxication. In our transcriptome analysis, we noticed an increase in the expression of specific genes related to oxidative stress in cells treated with the WT toxin when compared to those treated with the PltB^S35A mutant (Supplementary Fig. 12). Considering that typhoid toxin induces mitochondrial damage and dysfunction, we postulated that it might trigger the production of reactive oxygen species (ROS), known to impair mitochondrial functions and potentially lead to the release of mtDNA into the cytosol. To test this hypothesis, we used the cell-permeable fluorescent probe, 2',7'-dichlorodihydrofluorescein diacetate (CM-H₂DCFDA), a known indicator for ROS. Our observations detected a fluorescent change in WT toxin-treated cells as early as 8 h of treatment, confirming the presence of ROS in cells (Fig. 5a). We also assessed the production of mitochondrial ROS (mtROS) using fluorescence probes targeted specifically to the mitochondria (MitoSOX). Interestingly, our results revealed a significant increase in mtROS levels in cells exposed to WT typhoid toxin at a later time point (Fig. 5b). The variation in ROS detection at different time points with different probes could be attributed to two possible factors. First, it may be related to the probe specificity for ROS detection: CM-H₂DCFDA predominantly measures cellular $H_2O_2$, while MitoSOX provides an indication of mitochondrial superoxide levels. Second, there might be multiple factors contributing to ROS production from different organelles in cells, likely due to the influence of genomic DNA damage. To

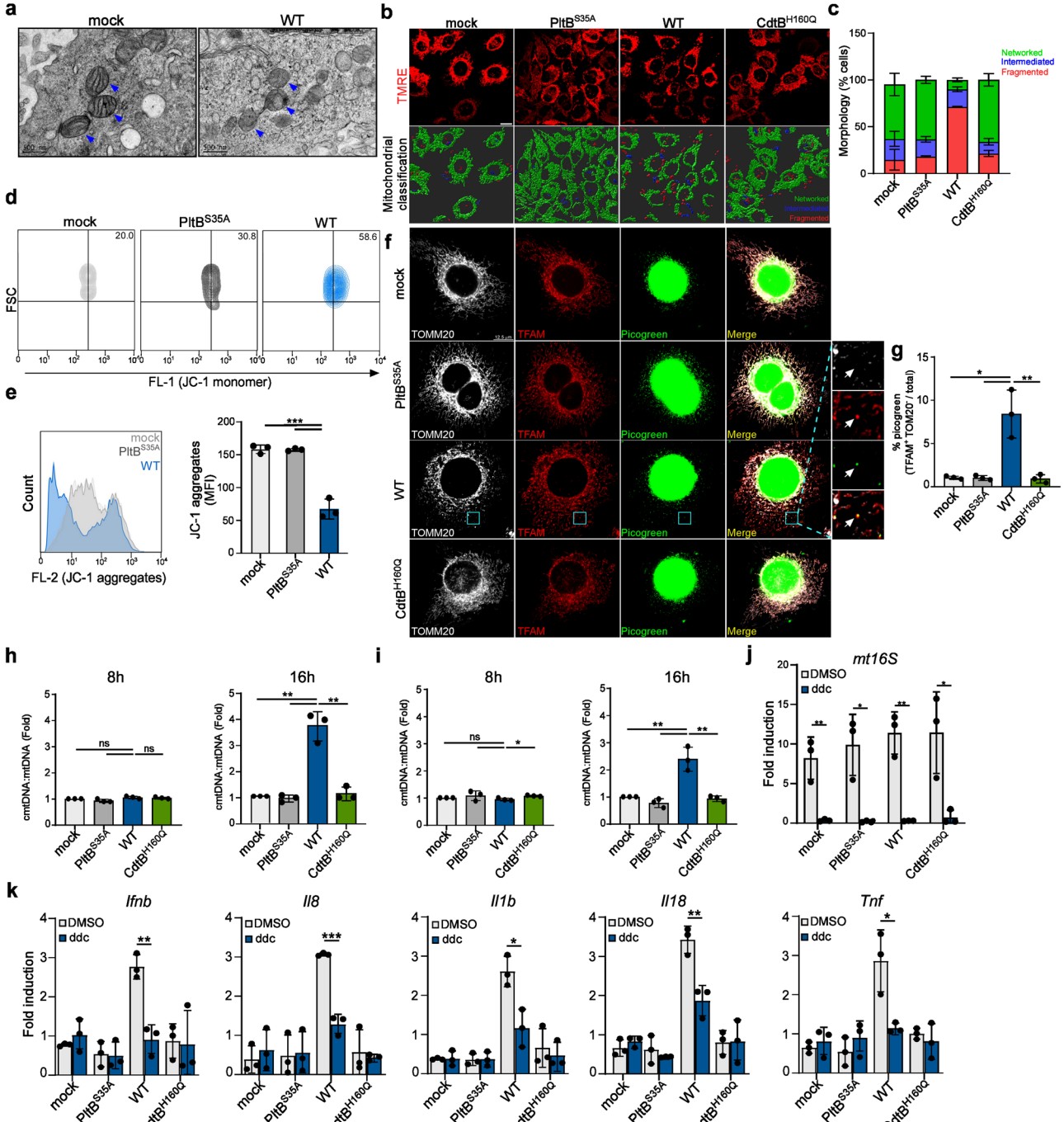

**Fig. 3 | Typhoid toxin activates the cGAS-STING signaling pathway by promoting mtDNA efflux into the cytosol. a**, Transmission electron microscopy of typhoid toxin-treated Henle-407 cells showing mitochondria (blue arrowhead). Scale bars, 500 nm. Cells are from 2 independent experiments. **b, c** Representative confocal images of TMRE staining in cells exposed to WT typhoid toxin and mutants. Scale bar, 15 μm (**b**). Mitochondrial morphology was classified into networked, intermediated, and fragmented categories using Imaris, with data representing the mean ± s.d (*n* = 3). At least 100 cells were counted for each condition (**c**). **d, e** Mitochondrial membrane potential measured by flow cytometry. THP-1-derived macrophages were treated with typhoid toxin. Sixteen hours after treatment, cells are stained with JC-1 dye (**d**). A histogram plot shows JC-1 intensity and quantification of JC-1 aggregate fluorescence indicating normal mitochondrial membrane potential. Data are presented as the mean ± s.d (*n* = 3) (**e**). Statistical analysis was performed using unpaired two-sided *t*-tests; ***P < 0.001. **f, g** Fluorescent images of DNA (green), TFAM (red) and mitochondria (white) in Henle-407 cells exposed to typhoid toxin. Scale bar, 12.5 μm. Magnified images

show the TFAM co-localizing with cytosolic DNA outside the mitochondrial network (**f**). Quantification of the ratio of cytosolic DNA foci distinct from those within the mitochondria relative to the total cytosolic DNA foci. Data are represented as the mean ± s.d. (*n* = 3), with 300 cells assessed for each condition (*t*-tests; *P < 0.05; **P < 0.01) (**g**). **h, i** RT-qPCR of cytosolic mtDNA (cmtDNA) relative to total mtDNA in THP-1-derived macrophages (**h**) and Henle-407 cells (**i**) exposed to typhoid toxin using the mt16S primer set. Data are presented as the mean ± s.d. (*n* = 3) with statistical significance (*t*-tests; *P < 0.05, **P < 0.01, ns, not significant). **j** RT-qPCR of total mtDNA in THP-1-derived macrophages exposed to typhoid toxin with or without ddC treatment. Mitochondrial DNA copy number was normalized by total nuclear DNA (*ACTB*). Data are presented as the mean ± s.d (*n* = 3) with statistical significance (*t*-tests; *P < 0.05, **P < 0.01). **k** RT-qPCR of indicated genes in THP-1-derived macrophages exposed to typhoid toxin in the presence of ddC treatment. Data are presented as the mean ± s.d (*n* = 3) with statistical significance (*t*-tests; *P < 0.05, **P < 0.01, ***P < 0.001). Source data are provided as a Source data file.

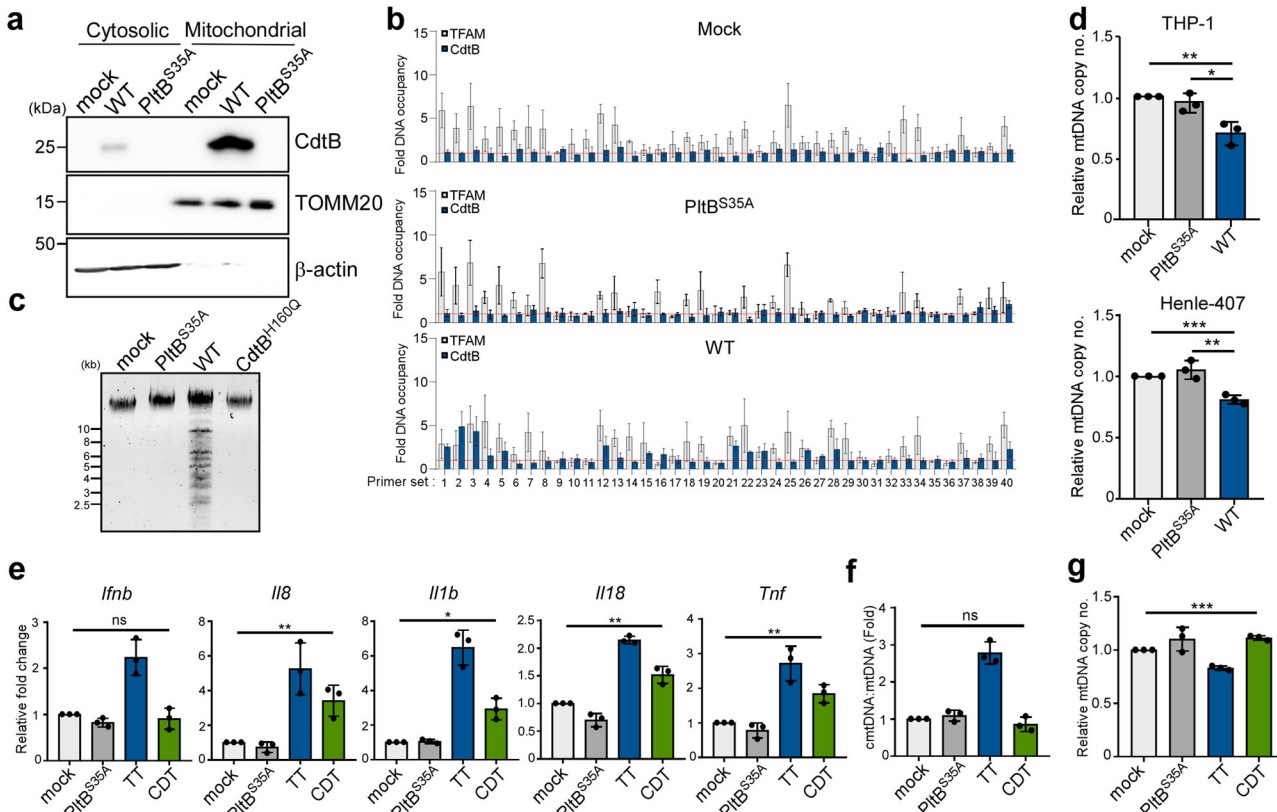

**Fig. 4 | Typhoid toxin interaction with mitochondrial DNA and damage induction. a** Presence of typhoid toxin in the mitochondrial fractions in the THP-1-derived macrophages. The cells were incubated with 0.2 μg of WT typhoid toxin and the PltB$^{S35A}$ mutant for 5 h and fractionated to determine the amount of typhoid toxin in the cytosolic and mitochondrial fractions by western blot analysis using antibodies against CdtB, TOMM20 (an outer mitochondrial membrane protein), and β-actin as a loading control. **b** Mitochondrial DNA immunoprecipitation (mtDIP) assay for typhoid toxin binding to mtDNA. CdtB and TFAM binding to mtDNA across the mitochondrial genome using 40 primer sets by RT-qPCR. Fold DNA occupancy calculated relative to the IgG-negative control ($n = 3$). **c** Agarose gel analysis of mtDNA samples following PCR amplification. Mitochondrial DNA, isolated from THP-1-derived macrophages exposed to either WT typhoid toxin or its mutants, was subjected to PCR amplification using primers targeting the entire mitochondrial genome. **d** RT-qPCR of mtDNA copy number normalized by total nuclear DNA (*ACTB*) in THP-1-derived macrophages and Henle-407 cells exposed to

WT typhoid toxin and the PltB$^{S35A}$ mutant version. Data are presented as the mean ± s.d ($n = 3$). Statistical analysis was performed using unpaired two-sided *t*-tests; *$P < 0.05$, **$P < 0.01$, ***$P < 0.001$. **e** RT-qPCR analysis of the mRNA levels in cells exposed to WT typhoid toxin, the PltB$^{S35A}$ mutant and the CDT. Statistical analysis was performed using unpaired two-sided *t*-tests ($n = 3$); *$P < 0.05$, **$P < 0.01$, ns, not significant ($P > 0.05$). **f** RT-qPCR of cytosolic mtDNA (cmtDNA) quantified relative to total mtDNA in THP-1-derived macrophages exposed to WT typhoid toxin (TT) (160 pM) and the CDT (60 nM). The primer set of *mt16S* was used to determine. Data are presented as the mean ± s.d ($n = 3$). Statistical analysis was performed using unpaired two-sided *t*-tests; ns, not significant ($P > 0.05$). **g** RT-qPCR of mtDNA copy number normalized by total nuclear DNA (*ACTB*) in THP-1-derived macrophages exposed to WT typhoid toxin and the CDT. Data are presented as the mean ± s.d ($n = 3$). Statistical analysis was performed using unpaired two-sided *t*-tests; ***$P < 0.001$. The western blots shown in (**a**) are representative of 3 independent experiments. Source data are provided as a Source data file.

assess the significance of ROS in the SASP, we exposed cells to typhoid toxin and subsequently treated them with ROS scavengers or a specific inhibitor, DPI, targeting NADPH oxidase (NOX), which is known for ROS production at the endosome or plasma membrane. Our results showed that the primary ROS scavenger GSH and the mitochondria-specific antioxidant, Mitoquinone (MitoQ), but not DPI, effectively inhibited the activation of the STING-mediated signaling pathway and reduced the mRNA levels of proinflammatory components (Fig. 5c–f, Supplementary Fig. 13). Furthermore, the administration of MitoQ decreased the induction of p16$^{INK4a}$ and SA-β-Gal activity (Fig. 5c, Supplementary Fig. 14). However, it did not affect the phosphorylation of p53 (Fig. 5c). This supports the notion that mtROS plays a role in inducing p16$^{INK4A}$-mediated senescence and the proinflammatory SASP response triggered by typhoid toxin.

We proceeded to investigate the interplay between mtROS and mtDNA leakage. Given the decrease in mtDNA copy number induced by typhoid toxin, we hypothesized that this reduction disrupts mitochondrial function, leading to an excessive generation of mtROS, subsequently causing the release of mtDNA into the cytosol (Fig. 5g). Remarkably, the use of MitoQ to neutralize mtROS

significantly mitigated the release of mtDNA into the cytosol (Fig. 5h), although it did not prevent the reduction in mtDNA copy number (Fig. 5i). This finding is consistent with our earlier observation of mitochondrial damage and suggests that the overproduction of mtROS results in damage to the mitochondrial membrane, leading to the escape of mtDNA into the cytosol. A recent study proposed that increased mtROS levels can lead to the association of gasdermin D (GSDMD), a pore-forming protein, with the mitochondrial membrane during inflammasome activation, culminating in mitochondrial pore formation and mtDNA release[25]. Although we did observe the upregulation of NLRP3 (although not AIM2 activation) and GSDMD cleavage induced by typhoid toxin (Supplementary Fig. 15a–d), pharmacological inhibition of GSDMD did not prevent the release of mtDNA into the cytosol or reduce IFN-β expression in typhoid toxin-treated cells (Supplementary Fig. 15e, f). These results imply the existence of an unidentified pathway for mtDNA release in the mtROS-mediated signaling mechanism. In summary, these findings establish a causal link between typhoid toxin-induced mtDNA damage, mtROS production, and mtDNA release, emphasizing the role of mtROS in facilitating the escape of mtDNA.

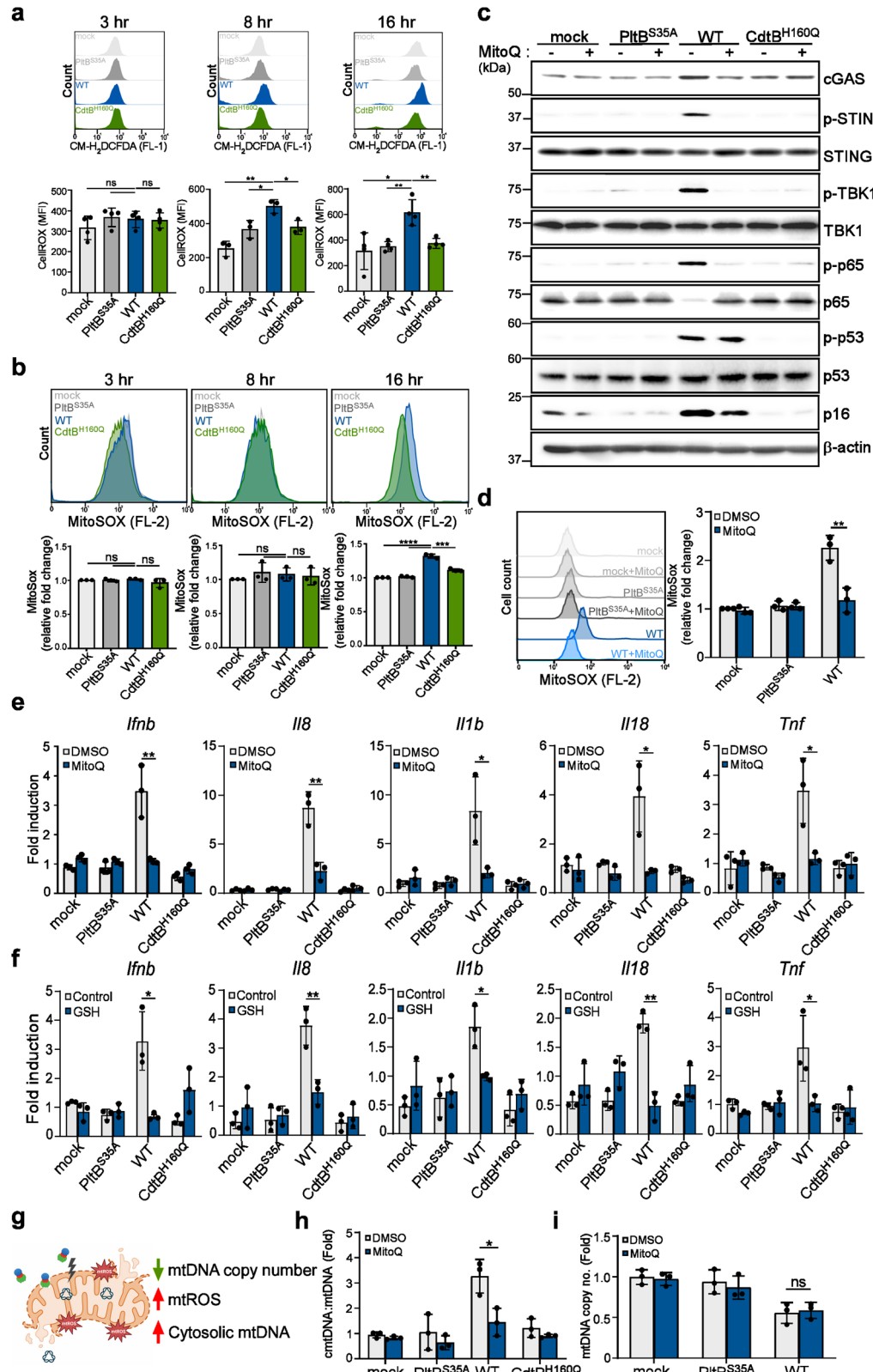

## GCN2-mediated ISR pathway contributes to the production of proinflammatory SASP factors

Our analysis of RNA-seq data also showed that cells exposed to WT typhoid toxin had a greater enrichment of genes associated with the ISR and mitogen-activated protein kinase cascade (MAPK) signaling pathway in comparison to those treated with the PltB[S35A] mutant (Fig. 6a, b). Activation of these pathways indicates a reprogramming

of gene expression that can drive the development of proinflammatory SASP[26–28]. The ISR signaling pathway can detect diverse stimuli, such as proteostasis stress, nutrient deprivation, viral infection, and heme deficiency. This pathway activates transcriptional rewiring, which induces proinflammatory responses that can restore cellular homeostasis[29,30]. MAPK (JNK, p38, and ERK1/2) are serine/threonine protein kinases that modulate proinflammatory responses

**Fig. 5 | Typhoid toxin-induced mitochondrial injury leads to the production of mitochondrial ROS to trigger the cytosolic release of mtDNA. a** Measurement of cellular ROS levels was performed in THP-1-derived macrophages exposed to WT typhoid toxin and different mutants. Mean fluorescence intensity (MFI) is presented as the mean ± s.d ($n$ = 3). Statistical analysis was performed using unpaired two-sided $t$-tests; *$P$ < 0.05, **$P$ < 0.01, ns, not significant ($P$ > 0.05). **b** Measurement of mitochondrial ROS was conducted in THP-1-derived macrophages exposed to typhoid toxin. Mean fluorescence intensity (MFI) is presented as the mean ± s.d ($n$ = 3). Statistical analysis was performed using unpaired two-sided $t$-tests; ***$P$ < 0.001, ****$P$ < 0.0001, ns, not significant ($P$ > 0.05). **c** THP-1-derived macrophages exposed to typhoid toxin and its mutants were incubated with MitoQ (2 μM) for 16 h. Cell lysates were analyzed using western blot with antibodies against cGAS, phosphorylated STING, phosphorylated TBK1, phosphorylated p65, phosphorylated p53, p16[INK4a], and β-actin as a loading control. **d** Mitochondrial ROS levels were assessed in THP-1-derived macrophages treated with typhoid toxin at 16 h post-treatment. Mean fluorescence intensity is presented as the mean ± s.d ($n$ = 3). Statistical analysis was performed using unpaired two-sided $t$-tests;

**$P$ < 0.01. **e, f** RT-qPCR analysis was conducted to measure the mRNA levels of the indicated genes in THP-1-derived macrophages exposed to WT typhoid toxin and different toxin mutants, with or without MitoQ (2 μM) (**e**) or GSH (10 mM) (**f**) treatment. Data are presented as the mean ± s.d ($n$ = 3). Statistical analysis was performed using unpaired two-sided $t$-tests; *$P$ < 0.05, **$P$ < 0.01. **g** The schematic model of mitochondrial damage induced by typhoid toxin (Created with BioRender.com). **h** RT-qPCR of cytosolic mtDNA (cmtDNA) quantified relative to total mtDNA in THP-1-derived macrophages exposed to typhoid toxin with MitoQ treatment. Data are presented as the mean ± s.d ($n$ = 3). Statistical analysis was performed using unpaired two-sided $t$-tests; *$P$ < 0.05. **i** RT-qPCR of mtDNA copy number in THP-1-derived macrophages exposed to WT typhoid toxin and the PltB[S35A] mutant with or without MitoQ treatment. Data are presented as the mean ± s.d ($n$ = 3). Statistical analysis was performed using unpaired two-sided $t$-tests; ns, not significant ($P$ > 0.05). The western blots shown in (**c**) are representative of 3 independent experiments. Source data are provided as a Source data file.

in conjunction with the NF-κB signaling pathway[31,32]. Our results showed that after 16 h of treatment with the WT toxin, the activation of p38 and JNK MAPKs, but not ERK MAPK, were noticeable in macrophages, as detected by western blot (Fig. 6c). However, inhibiting either p38 or JNK did not diminish the expression of proinflammatory components (Fig. 6d), suggesting that the MAPK signaling pathway does not seem to play a direct role in contributing to typhoid toxin-induced SASP.

We next examined the ISR signaling pathway by assessing the activation of eukaryotic translation initiation factor 2α (eIF2α) kinases and phosphorylation of eIF2α, a core event in the ISR. Specialized eIF2α kinases, including protein kinase R (PKR), general control non-repressed 2 (GCN2), protein kinase R-like ER kinase (PERK), and heme-regulated inhibitor (HRI), can respond to diverse stresses and converge on the phosphorylation of eIF2α (Fig. 6e)[29,30], which temporarily halts most protein synthesis but selectively increases the translation of certain genes. As HRI is mainly expressed in erythroid cells[33], we focused on other kinases. Our data indicated that only the amino acid sensor GCN2 kinase was activated, and eIF2α phosphorylation was evident in macrophages treated with WT typhoid toxin (Fig. 6f and Supplementary Fig. 16). Furthermore, we observed upregulation of ISR target genes such as activating transcription factor 3 (ATF3) and C/EBP homologous protein (CHOP) in cells exposed to the WT toxin (Fig. 6g, h). ATF3 serves as a crucial transcriptional regulator that can induce the expression of certain proinflammatory components[34,35]. Our findings showed that genetic ablation or pharmacological inhibition of GCN2 failed to induce eIF2α phosphorylation and upregulate ATF3 and CHOP expression in cells exposed to WT typhoid toxin (Fig. 6i,j and Supplementary Fig. 17 a, b), indicating GCN2 is responsible for the ISR triggered by typhoid toxin. We also found that disruption of the ISR pathway decreased the mRNA levels of proinflammatory components but did not affect the expression of *Ifnb* (Supplementary Fig. 17c and Fig. 6k, l). Interestingly, the GCN2-mediated ISR pathway was inactive in STING-deficient cells (Fig. 6m, and Supplementary Fig. 18). Taken together, these results indicate a pivotal role of the GCN2-mediated ISR pathway in the development of typhoid toxin-induced SASP, and its activation is dependent on the STING signaling pathway.

### Typhoid toxin-induced SASP in senescent macrophages promotes phenotypic characteristics of senescence in T cells

Recent research has unveiled the potent influence of the SASP on immune regulation, orchestrating immune cell recruitment, and transmitting the SASP signal to neighboring cells via the paracrine pathway[17,36]. We next explored the impact of the secretomes of senescent macrophages induced by typhoid toxin on T cells, as these cells play a pivotal role in immune-mediated senescence surveillance

and serve as a bridge between the innate and adaptive immune systems in controlling *Salmonella* infection[37–39]. We discovered that exposing CD4 T cells to culture supernatants (conditioned medium) from macrophages treated with WT typhoid toxin resulted in a G2/M phase cell cycle arrest (Supplementary Fig. 19a) and inhibited proliferation 6 days after incubation (Fig. 7a, and Supplementary Fig. 19b). In addition, conditioned medium from WT toxin-treated macrophages led to a substantial increase in cell death (Fig. 7b), accompanied by elevated SA-β-gal activity in activated CD4 T cells (Supplementary Fig. 19c). Furthermore, CD4 T cells exposed to conditioned medium from WT toxin-treated cells exhibited increased protein levels of killer cell lectin-like receptor G1 (KLRG1) and T cell immunoreceptor with Ig, ITIM domains (TIGIT), along with a decrease in the co-stimulatory molecule CD28 (Fig. 7c–e). These CD4 T cells also displayed an upregulation of granzyme B (GzmB) and increased secretion of IFN-γ (Fig. 7f, g). These biomarkers collectively indicate the induction of senescence in CD4 T cells, which can negatively impact immune surveillance[40]. In a similar vein, exposure of CD8 T cells to conditioned medium also induced phenotypic traits of senescence in activated CD8 T cells (Supplementary Fig. 19d–g).

We then determined whether this senescent bystander effect on T cells is triggered by mtDNA-associated senescence within typhoid toxin-treated macrophages. To delve into this, activated CD4 T cells were exposed to conditioned medium obtained from typhoid toxin-treated macrophages in which mtDNA had been depleted using ddC. We observed increased CD4 T cell proliferation and the restoration of normal cell cycle regulation (Fig. 7h, i). Furthermore, this conditioned medium was able to rescue cell death and reduce the protein expression of biomarkers indicative of T cell senescence (Fig. 7j–o). Consistent with this, the conditioned medium from macrophages deficient in STING had similar effects on T cells as mtDNA depletion (Supplementary Figs. 20 and 21), further emphasizing the critical role of typhoid toxin in mtDNA-associated SASP. The alterations in the expression levels of these senescence markers in activated T cells collectively suggest that SASP factors secreted by typhoid toxin-treated cells can indeed induce a bystander senescence effect on neighboring cells.

## Discussion
Our findings reveal that typhoid toxin disrupts mitochondrial function, setting off a sequence of events that ultimately trigger the SASP. Specifically, we demonstrate that typhoid toxin induces damage to mitochondrial DNA, resulting in mitochondrial oxidative stress, which leads to the release of mtDNA into the cytosol. This release then activates the cGAS-STING pathway and the ISR, causing changes in the host cell transcriptome and the production of proinflammatory SASP

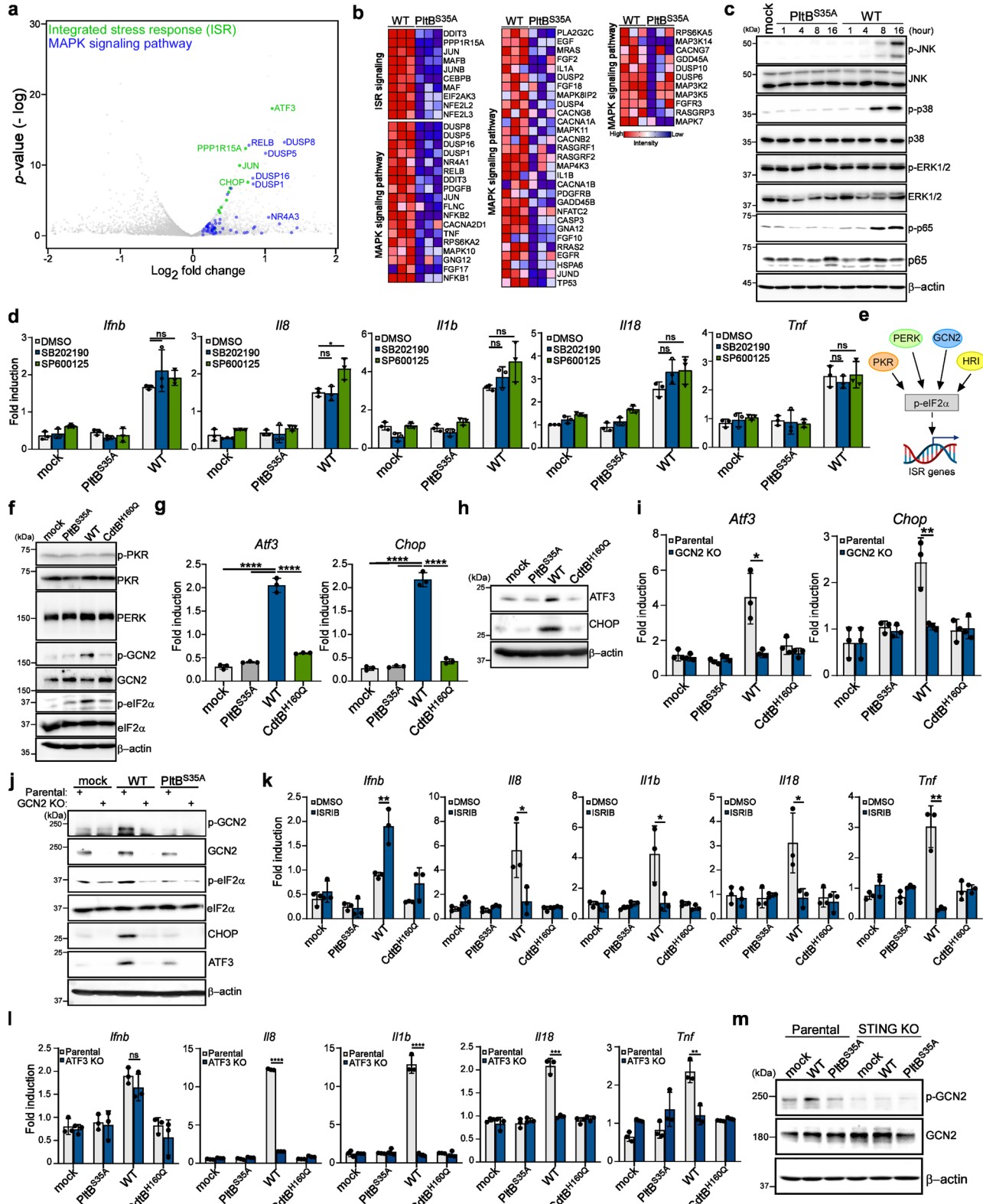

components. Ultimately, these SASP factors have the ability to induce paracrine senescence in T cells, influencing their fate and effector functions (Fig. 8). Considering that typhoid fever is characterized by intense and prolonged inflammatory responses, which can lead to complications such as gut perforation and disruption of tissue homeostasis, our study highlights the physiological impact of typhoid toxin by linking senescence-associated immune responses with mitochondrial damage caused by typhoid toxin. This intricate interplay

between the toxin, cellular senescence, and mitochondrial dysfunction may contribute to the pathogenesis of typhoid fever.

Unlike DNA-damaging agents like hydroxyurea[41,42], which does not affect target cells in the same manner as typhoid toxin (Supplementary Fig. 22), the unique action of typhoid toxin in damaging mitochondria and inducing the expression of proinflammatory components highlights its toxin-specific nature. These effects are a direct result of the distinct properties of typhoid toxin. Moreover, in contrast to other

**Fig. 6 | GCN2-mediated integrated stress response contributes to the production of proinflammatory SASP components. a** Volcano plot showing upregulated genes related to the ISR signaling (green) and mitogen-activated protein kinase cascade (MAPK) signaling pathway (blue) in WT typhoid toxin-treated macrophages compared to PltB$^{S35A}$ mutant toxin-treated macrophages using an arbitrary threshold of $P < 0.05$ and fold change >1.5. **b** Expression of genes involved in the ISR signaling and MAPK signaling pathway by RNA-seq. **c** Immunoblot analyses demonstrate the activation of MAPK signaling pathways in THP-1-derived macrophages treated with typhoid toxin. Phosphorylated JNK, p38, ERK1/2, p65, and β-actin (loading control) were monitored at various time points. **d** THP-1-derived macrophages exposed to typhoid toxin were treated with p38 MAPK (SB202190) (1 μM) and JNK MAPK (SP600125) (5 μM) inhibitors for 16 h. Total mRNA was analyzed using RT-qPCR. Data are presented as mean ± s.d ($n = 3$) ($t$-tests; *$P < 0.05$, ns, not significant). **e** A model (Created with BioRender.com) of the ISR signaling. **f** The ISR signaling pathway in THP-1-derived macrophages exposed to typhoid toxin was analyzed using western blot analysis with specific antibodies. **g**, **h** Total mRNA and

protein levels of ISR target genes were quantified using RT-qPCR (**g**) and western blot analysis (**h**), respectively. Data are presented as mean ± s.d ($n = 3$) with statistical significance ($t$-tests; ****$P < 0.0001$). **i** The mRNA levels of specific genes were assessed using RT-qPCR. Data are presented as mean ± s.d ($n = 3$). Statistical analysis was performed using unpaired two-sided $t$-tests; *$P < 0.05$, **$P < 0.01$. **j** Western blot analyses were performed to assess the activation of the GCN2-mediated ISR pathway. **k, l** The mRNA levels of proinflammatory genes were measured in THP-1-derived macrophages exposed to typhoid toxin under ISRIB treatment (30 μM) (**k**) or in ATF3-deficient cells and their parental cell line (**l**). RT-qPCR was used for gene expression analysis. Data are presented as mean ± s.d ($n = 3$) ($t$-tests; *$P < 0.05$, **$P < 0.01$, ***$P < 0.001$, ****$P < 0.0001$, ns, not significant.). **m** Activation of GCN2 in STING-deficient THP-1-derived macrophages and their parental cell line exposed to typhoid toxin was determined by western blot analysis. The western blots shown in (**c**), (**f**), (**h**), (**j**), (**m**) are representative of 3 independent experiments. Source data are provided as a Source data file.

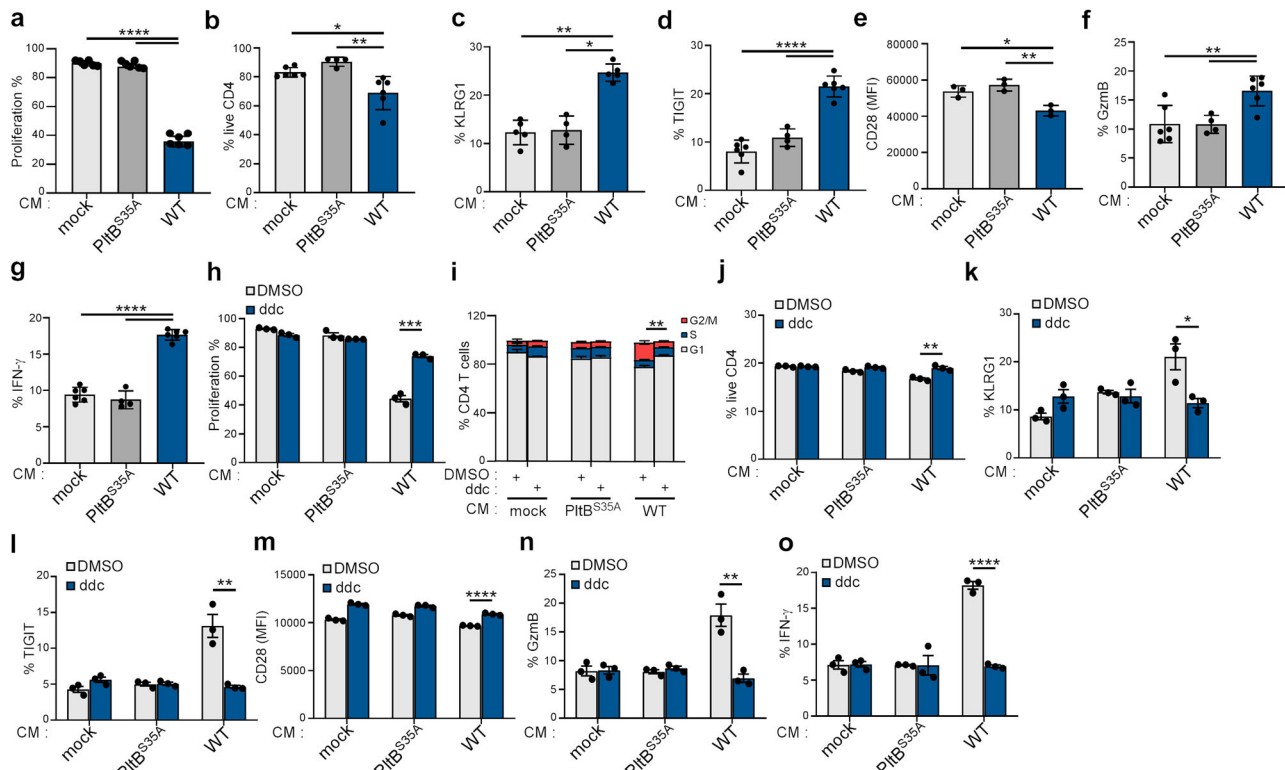

**Fig. 7 | Typhoid toxin-induced SASP in senescent macrophages promotes phenotypic characteristics of senescence in T cells. a** Total T cells isolated from wild-type mice were activated with anti-CD3/CD28, labeled with CellTrace Violet, and cultured in supernatants from RAW 264.7 macrophages treated with mock, wild-type (WT) typhoid toxin, or the PltB$^{S35A}$ mutant. The percentages indicate the proportion of cells that exhibited proliferation 6 days. **b–g** Murine primary T cells were activated with anti-CD3/CD28 and then restimulated with ionomycin/TPA for 5 h, and the characteristics of cells were determined by co-staining of indicated surface and intracellular proteins. Frequency of live CD4$^+$ T cells (CD4$^+$) (**b**). Frequency of cell surface markers KLRG1 (CD4$^+$KLRG1$^+$) and TIGIT (CD4$^+$TIGIT$^+$) in CD4 T cells (**c, d**). Expression of CD28 in CD4$^+$ T cells (CD4$^+$CD28$^+$) (**e**) determined by

mean fluorescence intensities (MFI). Frequency of granzyme B (CD4$^+$GzmB$^+$) (**f**) and IFN-γ (CD4$^+$IFN-γ$^+$) (**g**) positive cells in CD4 T cells. Data are presented as the mean ± s.d ($n = 3$). Statistical analysis was performed using unpaired two-sided $t$-tests; *$P < 0.05$, **$P < 0.01$, ****$P < 0.0001$. **h–o,** T cells activated as previously described were cultured in supernatants (CM) from RAW 264.7 macrophages treated with mock, WT typhoid toxin, or the PltB$^{S35A}$ mutant, with or without ddC. T cell proliferation (**h**), cell cycle (**i**), cell viability (**j**), and the expression of indicated surface and intracellular proteins (**k–o**) in CD4 T cells were analyzed using flow cytometry. Data are presented as the mean ± s.d ($n = 3$). Statistical analysis was performed using unpaired two-sided $t$-tests; *$P < 0.05$, **$P < 0.01$, ****$P < 0.0001$. Source data are provided as a Source data file.

genotoxins such as CDT, which target nuclear genomic DNA to induce cellular senescence via the ATM-p38 signaling axis, typhoid toxin's surprising strategy of targeting mitochondria stands out. Our study reveals that exposure to typhoid toxin results in a reduction in mtDNA copy number due to its direct interaction with mtDNA. Intriguingly, we found that the CdtB subunit of typhoid toxin exhibits a strong preference for specific regions within the mitochondrial genome. Its highest binding affinity lies within the D-loop region and the segment

from the light-strand promoter (LSP) to the heavy-strand promoters (HSP1 and HSP2). Although the D-loop is non-coding, it plays a critical role in mitochondrial genome replication and protein expression[43]. The LSP and HSP promoters are recognized by TFAM[44,45], a vital transcriptional factor crucial for maintaining the integrity of the mitochondrial genome. This interaction between typhoid toxin and these regions raises the possibility that typhoid toxin has a profound impact on mtDNA, leading to mitochondrial genome instability. Furthermore,

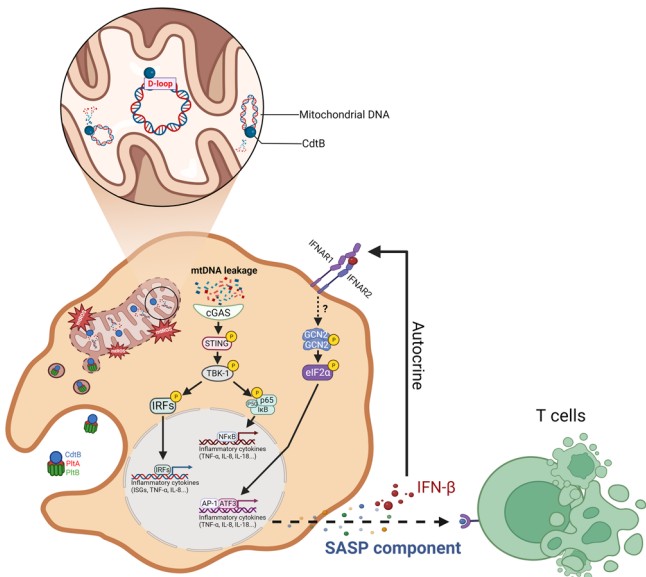

**Fig. 8 | The model for the induction of proinflammatory SASP by typhoid toxin.** Typhoid toxin's mechanism of action involves its entry into the mitochondria, where it initiates the proinflammatory SASP by directly targeting mitochondrial DNA (mtDNA) and inducing damage to this essential genetic material. This mtDNA damage triggers a cascade of events, beginning with the disruption of mtDNA integrity, which subsequently leads to mitochondrial dysfunction and a disturbance in redox homeostasis. As a consequence of these perturbations, damaged mtDNA is released into the cytosol, activating the cGAS-STING signaling pathway. This activation, in turn, instigates the expression of proinflammatory components, ultimately contributing to the development of the proinflammatory SASP (Created with BioRender.com).

CdtB's second preferred binding site spans nucleotide positions 8459 to 11802 and encodes crucial proteins involved in the electron transport chain, including MT-ATP8, MT-ATP6, MT-CO3, MT-ND4L, and MT-ND4. Damage to these DNA regions can disrupt mitochondrial functions, resulting in mitochondrial injury. The consequences of the toxin's actions not only lead to the release of mtDNA into the cytosol but also activate caspase-9 (Supplementary Fig. 15a), triggered by the release of cytochrome c from the damaged mitochondria.

Our study reveals that typhoid toxin activates the GCN2-mediated integrated stress response (ISR), contributing to the production of SASP components. Interestingly, our results show the dependency of GCN2-mediated ISR on STING signaling in cells exposed to typhoid toxin. However, the precise mechanisms underlying how STING signaling regulates GCN2-mediated ISR have yet to be discovered. GCN2, known for sensing amino acid deprivation, modulates protein synthesis in response to nutrient scarcity. We observed an upregulation in the mRNA level encoding IDO1 (Supplementary Fig. 23), an enzyme responsible for metabolizing the essential amino acid tryptophan and rapidly depleting the intracellular tryptophan pool. Given that IDO1 is an interferon-stimulated gene[46], we postulate that typhoid toxin-induced type I IFN production via the STING signaling pathway stimulates the expression of IDO1, leading to tryptophan depletion and subsequent activation of GCN2. This cascade of events triggers a secondary immune response.

In addition to the cellular signaling pathways analyzed in our RNA-seq data, we have identified several SASP-related pathways. The PPAR signaling pathway is known for its role as transcription factors that govern gene expression related to various pathways including lipid metabolism and inflammation. Interestingly, PPAR transcriptional activity can be influenced by MAP kinases, as demonstrated in the present study. Moreover, it is worth noting that the PPAR signaling pathway can respond to DNA damage and has been documented to promote cellular senescence by inducing the expression of p16[INK4a], suggesting its pivotal role in the regulation of senescence[47]. Another identified pathway is SNARE interactions in vesicular transport. SNARE proteins are responsible for driving membrane fusion in vesicle trafficking[48]. The coordinated efforts of SNARE proteins are crucial for the exocytic pathway, which is essential for the secretion of SASP components in senescent cells[49,50]. Moreover, data from the SASPAtlas database (saspatlas.com) indicate an augmentation in the production of extracellular vesicles (EVs) within the SASP[51]. These EVs can serve as pivotal mediators in cell-cell communication during cellular senescence.

We also observed the upregulation of genes associated with focal adhesion and adherens junction (AJ) pathways in cells exposed to typhoid toxin. Senescence-associated morphological changes represent a prominent feature of senescent cells. These changes manifest as a flattened, enlarged cell shape, accompanied by an increase in the size and composition of focal adhesions. This includes enhanced expression of integrins and increased activity of the focal adhesion complex (FAK). Furthermore, our RNA-seq results, along with a study conducted by ElGhazaly et al., both identified an upregulation of the senescence-associated secretory phenotype (SASP) factor, transforming growth factor-β (TGF-β), in senescent cells exposed to typhoid toxin[18]. Elevated TGF-β levels can activate FAK and influence cell-cell adhesion[52], thereby regulating senescence. Adherens junctions have a crucial role in facilitating cell-cell adhesive interactions, contributing to the remodeling of the actin cytoskeleton and significantly impacting tissue organization[53]. In addition, Wnt signaling, stimulated by TGF-β, is involved in AJ regulation[54–57]. Recent research has shed light on the role of AJs in the formation of senescent cell adhesion fragments (SCAFs), present in various types of senescent cells[58]. These SCAFs act as damage-associated molecular patterns (DAMPs) and play a role in wound healing processes. While these findings from pathway analysis are intriguing, further studies are needed to elucidate their significance and implications.

Recent studies have highlighted the shared phenotypic characteristics between macrophages and senescent cells[59], such as heightened immune responses and increased protein secretion, which can resemble a senescent-like phenotype. Furthermore, the detection of p16[INK4a] expression and elevated SA-β-gal activity in macrophages has been noted as a physiological response to immune stimuli[60]. These observations highlight the similarities between macrophage senescence and immune activation, necessitating further investigation into macrophage biology. Despite these limitations, our study provides insights into transcriptomic changes in macrophages following typhoid toxin treatment, suggesting a state of cellular senescence alongside induction of common senescence biomarkers. These findings contribute to understanding the genotoxin's impact on macrophages, their senescence-related activity and inflammatory responses. Overall, our research elucidates the mechanisms underlying typhoid toxin actions and their role in exacerbating bacterial infection pathologies. This advancement deepens our understanding of the complex interplay between hosts and pathogens, paving the way for potential therapeutic interventions.

## Methods
### Plasmid, antibodies, and reagents
All the plasmids listed in Supplementary Data 1 were constructed using the Gibson assembly cloning strategy and verified by nucleotide sequencing. Antibodies and reagents used in the study are listed in Supplementary Data 2 and purchased from the indicated commercial sources.

### Cell culture
THP-1 cells obtained from the American Type Culture Collection were cultured in Roswell Park Memorial Institute (RPMI)-1640 supplemented

with 10% heat-inactivated fetal bovine serum (FBS), 4.5 g/L glucose, 10 mM HEPES, and 10 mM sodium pyruvate. THP-1 cells were differentiated into macrophages by treating cells with 40 nM of phorbol 12-myristate 13-acetate (PMA) for 48 h, followed by overnight incubation in fresh medium before experiments. To deplete mitochondrial DNA, THP-1-derived macrophages were incubated in fresh medium with 20 µM ddC for 24 h before experiments. Human intestinal epithelial Henle-407 cells (obtained from Jorge Galan's laboratory at Yale University) and HEK293T cells (from the American Type Culture Collection) were cultured in DMEM supplemented with 10% FBS. All cell lines were cultured at 37 °C in a humidified culture incubator with 5% CO2 and routinely screened for mycoplasma tests using a standard PCR method.

### Bone marrow-derived macrophage culture and differentiation
Bone marrow-derived macrophages (BMDMs) were cultured from 6- to 8-week-old C57BL/6 mice using a previously established protocol. In brief, BMDMs were grown in completed DMEM medium supplemented with 10% heat-inactivated fetal bovine serum (FBS) and 10 ng/ml of recombinant M-CSF (PeproTech). The undifferentiated M0 macrophages were maintained for 7 days. To induce differentiation into distinct macrophage subsets, the M0 macrophages were exposed to specific stimuli. Proinflammatory M1 macrophages were generated by incubating M0 macrophages with 10 ng/ml of IFN-γ (R&D Systems) and 100 ng/ml LPS (Sigma-Aldrich) for 24 h. Alternatively, incubation with 20 ng/ml of IL-4 (PeproTech) and IL-13 (PeproTech) for 24 h prompted the differentiation of alternative M2 macrophages.

### Generation of the Cas9 stable cell line
To establish cell lines with stably expressing wild-type Cas9 endonuclease (THP-1-Cas9), THP-1 cells were transduced with lentiviral particles produced from lentiCas9-Blast (Addgene, #52962) and subsequently selected for resistance to blasticidin.

### LentiCRISPR virus production and CRISPR/Cas9 gene inactivation in cultured cells
HEK293T cells were seeded on 60 mm tissue culture dishes and allowed to grow to 30–40% confluence. Each well was then transfected with pLentiGuide-Puro carrying sgRNA, pVSVg, and psPAX2 plasmid DNA using Lipofectamine™ 3000 Transfection Reagent (Life Technologies). After a 4-h incubation, the media was changed with DMEM supplemented with 10% FBS and 1% PS. Three days after transfection, the culture media containing the viruses for lentiviral virus transduction on the target cell line (THP-1 cells) was pooled and centrifuged at 3000 rpm for 10 min at 4 °C to pellet cell debris. The supernatants were filtered through 0.22 µm low-protein-binding membranes. The transduced cells were selected in the medium with puromycin (10 µg/ml) for 7 days, and isolated clones were further screened by immunoblot to identify knockout cells. Two independently isolated clones per cell line were characterized for the relevant phenotypes in the study, and in all cases, the different cell lines exhibited equivalent phenotypes.

### Typhoid toxin purification and expression
Purification of typhoid toxin and cytolethal distending toxin (CDT) was carried out as described previously[1]. In brief, the genes encoding typhoid toxin in *Salmonella* Typhi (*pltB*/*pltA*/6xHis-*cdtB*) or CDT in *Campylobacter jejuni* (*cdtA*/6xHis-*cdtB*/*cdtC*) were cloned into the pET28a expression vector. The *Escherichia coli* strain BL21 (DE3) carrying the different plasmids was grown in LB media at 37 °C to an OD600 of ~0.6. To induce toxin expression, 0.5 mM IPTG was added, followed by overnight incubation at 25 °C for typhoid toxin or 5 h at 37 °C for CDT. Bacterial cell pellets were resuspended in a buffer containing 15 mM Tris-HCl (pH 8.0), 150 mM NaCl, 0.1 mg/ml DNase, 0.1 mg/ml lysozyme, and 0.1% PMSF and lysed using a sonicator.

Toxins were then purified from bacterial cell lysates using affinity chromatography on a Nickel-resin (Bio-Rad), ion exchange, and gel filtration as previously described[1]. Purity of purified toxins was examined by SDS-PAGE gels stained with Coomassie blue.

### Immunofluorescence staining of cytosolic double-stranded DNA (dsDNA) in cells
Henle-407 cells were treated with different versions of 160 pM of typhoid toxin at 37 °C for 60 min and subsequently switched to regular growth medium. After 24 h, the cells were fixed with 4% paraformaldehyde for 15 min at room temperature, followed by washing with DPBS three times. The cells were then permeabilized with 0.1% Triton X-100 in DPBS for 20 min at room temperature and stained overnight at 4 °C with antibodies against dsDNA (Abcam) and the Alexa 488 conjugated antibody (Invitrogen) for 1 h at room temperature. The cells were then imaged and visualized using the EVOS M7000 microscope.

### Immunofluorescent staining of cytosolic mitochondrial DNA in cells
Henle-407 cells were initially treated with 160 pM of typhoid toxin at 37 °C for 60 min and then transitioned to regular growth medium. After 24 h, the cells were subjected to a 2-h incubation with PicoGreen (Thermo Fisher) at a 1:300 dilution. Subsequently, they were fixed with 4% paraformaldehyde at room temperature for 15 min and rinsed three times with DPBS. The cells were then permeabilized with 0.1% Triton X-100 in DPBS for 20 min at room temperature and stained overnight at 4 °C with primary antibodies against TFAM (Cell Signaling) and TOMM20 (Santa Cruz). Subsequent staining was conducted with an overnight incubation using a goat anti-rabbit IgG cross-absorbed secondary antibody, biotin-XX (Invitrogen), at 4 °C. The following day, the cells were stained with Alexa Fluor™ 568 streptavidin and Alexa Fluor 647-conjugated antibodies (Invitrogen) for 1 h at room temperature. The cells were imaged and visualized using the EVOS M7000 microscope.

### RNA preparation and RT-qPCR
Total RNAs from mammalian cells were extracted with Nucleozol reagent (Machnery-Nagel). Complementary DNA was synthesized using the PrimeScript RT Reagent Kit with gDNA Eraser (Takara Bio) according to the manufacturer's instructions. RT-qPCR was performed on a BioRad CFX96 machine using qPCRBIO SyGreen Blue Mix (PCR Biosystems) to quantify the expression of the target genes. The expression levels were normalized to β-actin for each sample. The primers used for qRT-PCR are listed in Supplementary Data 3.

### RNA sequencing and bioinformatic analysis
THP-1-derived macrophages were exposed to 160 pM of typhoid toxin at 37 °C for 60 min and then changed to regular growth medium for 16 h. The following day, total RNAs from the cells were isolated using RNeasy purification kits (QIAGEN) following the manufacturer's instructions. The extracted RNA was then converted into cDNA with the Illumina Stranded mRNA Prep kit. The resulting samples were sequenced on the HiSeq NovaSeq 6000 (Illumina) at the Medical Microbiota Center of the First Core Laboratory in the College of Medicine, National Taiwan University. Data processing and analyses were performed using QIAGEN CLC Genomics Workbench (v.21), followed by functional analyses conducted with GSEA in conjunction with MSigDB (v.7.0).

### Flow cytometry assessment of TLR9 expression
To perform intracellular staining of TLR9, the cells were kept at 4 °C and treated with an anti-mouse CD16/32 antibody (BioLegend) to prevent non-specific binding. After a 20-min fixation period, the cells were permeabilized with 0.1% saponin and subsequently subjected to

staining using the FITC anti-mouse CD289 (TLR9) Antibody (clone S18025A, BioLegend) for 1 h. The analysis of these stained cells was conducted using flow cytometry.

## SA-β-galactosidase staining

Cells were initially incubated with 160 ρM of typhoid toxin at 37 °C for 60 min and then replaced with regular culture medium. After 48 h, cells were rinsed with DPBS and fixed for 10 min in a solution containing 2% formaldehyde and 0.2% glutaraldehyde. Subsequently, the fixed cells were incubated with a staining solution containing 40 mM citrate, 5 mM potassium ferrocyanide, 2 mM MgCl$_2$ with 1 mg/ml of 5-bromo-4-chloro-3-indolyl-β-D-galactoside (X-Gal) at 37 °C for 16 h. Cells were then observed and images were acquired using the EVOS M7000 fluorescence microscope. For each sample, a minimum of 100 cells were counted in randomly selected fields.

## ELISA

THP-1-derived macrophages ($8 \times 10^5$) were treated with 160 ρM of typhoid toxin at 37 °C for 60 min followed by replacing with regular culture medium. Twenty-four hours following typhoid toxin treatment, cell lysates were collected and analyzed using ELISA, following the instructions provided by the human IL-8 ELISA, IFN-β, and TNF-α kits.

## Cytokine array

The cytokine array (Abcam) was performed in accordance with the manufacturer's guidelines for the Human Cytokine Antibody Array (Abcam). In brief, cell lysates were obtained from typhoid toxin-treated THP-1-derived macrophages ($3 \times 10^6$) and then subjected to cytokine array after a 1:5 dilution. The diluted samples were incubated with the dot blot overnight at 4 °C. The following day, the dot blots were thoroughly with a wash buffer, followed by overnight incubation with the cytokine biotin-conjugate antibody at 4 °C. The dot blots were then probed with HRP Streptavidin for 1 h at room temperature and subsequently detected using the Western Lightning Pro ECL detection reagent (PerkinElmer).

## Cell cycle analysis

Cell cycle analysis was assessed by flow cytometry following a previously described protocol[11,13]. In brief, cells were collected and fixed overnight with 70% ethanol in DPBS at −20 °C. Fixed cells were washed with DPBS, resuspended in 1 ml of DPBS containing 50 μg/ml propidium iodide, 0.1 mg/ml RNase A and 0.5% Triton X-100, and incubated for 30 min at 37 °C. After washing with DPBS and filtering, cells were analyzed by flow cytometry using a BD FACS Calibur Flow Cytometer. The DNA content of cells was determined with FlowJo (v10.8.1).

## Luciferase reporter assay

Indicated cell lines were transfected with plasmids expressing pIFNβ-mediated firefly luciferase and the control pCMV-Renilla luciferase reporter. Following an overnight incubation, the cells were exposed to 160 ρM of typhoid toxin at 37 °C for 60 min and then changed to regular growth medium. The following day, cell lysis was carried out using 1XPLB solution (Promega). The activity of IFNβ promoter and Renilla luciferase were measured using the SpectraMax i3x Multi-Mode Microplate Reader (Molecular Devices). The data obtained from firefly and Renilla luciferase activities were normalized by calculating the ratio of firefly and Renilla luciferase activity to determine the normalized IFNβ promoter activity.

## Western blot analysis

Protein samples lysed with lysis buffer (0.5% Triton X-100, 150 mM NaCl, 50 mM Tris-HCl, and protease inhibitors) were subjected to analysis using SDS-PAGE gels and subsequently transferred onto a nitrocellulose membrane. The primary antibodies are listed in Supplementary Data 2. The nitrocellulose-bound primary antibodies were detected using anti-mouse/rabbit IgG horseradish peroxidase-linked antibodies (Jackson ImmunoResearch). The detection was performed using the Western Lightning Pro ECL detection reagent (PerkinElmer).

## Detection of cytosolic and mitochondrial reactive oxygen species

The cells were treated with 160 ρM of typhoid toxin at 37 °C for 60 min and then changed to regular growth medium. At indicated time points, the cells were incubated with a CM-H$_2$DCFDA (5-(and-6)-chloromethyl-2′7′-dichlorodihydrofluorescein diacetate acetyl ester; Abcam) for 45 min at 37 °C, and then washed with DPBS. The stained cells were harvested by trypsinization and resuspended in DPBS. ROS detection was assessed by flow cytometry, and results were analyzed with the FlowJo software. To detect mitochondrial ROS (mtROS), cells were treated with MitoSOX (Invitrogen) for 30 min following the manufacturer's instructions. After washing with HBSS twice, cells were analyzed using flow cytometry.

## Mitochondria isolation and proteinase K sensitivity assay

Mitochondria were isolated using the mitochondria isolation kit (Invitrogen) following the manufacturer's instructions. The purity of mitochondrial and cytosolic fractions was analyzed by immunoblotting. The extracted mitochondria were then incubated with 12.5 μg/ml proteinase K in the presence or absence of 1% Triton X-100 for 30 min at 37 °C. To terminate the reaction, the samples were treated with 5 μM phenylmethanesulfonylfluoride (PMSF) for 5 min at room temperature to inactive proteinase K. The samples were then analyzed by SDS-PAGE and immunoblot using antibodies against typhoid toxin, TOMM20, and COX4.

## Measurement of cytosolic mtDNA

Cytosolic mtDNA was analyzed by RT-qPCR as previously described. In brief, the cells were cytoplasmically lysed in digitonin buffer (150 mM NaCl, 50 mM HEPES, pH 7.4, 25 μg/ml digitonin) with rotation for 10 min at room temperature followed by centrifugation at $16,000 \times g$ for 25 min at 4 °C. Cytosolic mtDNA (cmtDNA) was obtained from the supernatant cytosolic fraction, and total mtDNA was isolated from the whole-cell lysate using a genomic DNA isolation kit. The DNA solution was diluted 10-fold with nuclease-free deionized distilled H$_2$O and subjected to RT-qPCR. The level of cmtDNA was normalized by the mtDNA in the total lysates. The primer sequences are listed in Supplementary Data 3.

## Mitochondrial DNA damage assessment by long-range PCR

Long-range PCR was performed utilizing Expand Long Template PCR System (Roche), with amplification focused on mitochondrial DNA through a pair of primers specially tailored for this purpose. Comprehensive primer sequences can be referenced in Supplementary Data 3. To prepare the LA Taq reaction for all long-range PCR reactions (50 μl each), the following components were used: 5 μl of buffer, 0.35 μM of dNTPs, 0.35 μM of both forward and reverse primers, 3.75 units of polymerase, 5 μl of DNA (250 ng), and nuclease-free water was added to reach a total volume of 50 μl. The initial cycling conditions included an initial denaturation at 94 °C for 1 min, followed by 10 cycles of denaturation at 94 °C for 2 min, annealing at 55 °C for 30 s, and extension at 68 °C for 12 min. The subsequent cycling conditions consisted of 10 cycles with denaturation at 94 °C for 15 min, annealing at 55 °C for 30 s, and extension at 68 °C for 20 min. Finally, there was a further extension step at 72 °C for 7 min. The resulting PCR products were subsequently separated and visualized via 0.7% agarose gel electrophoresis.

## Mitochondrial DNA immunoprecipitation (mtDIP)

The mtDIP assay was conducted following established procedures. In brief, mitochondria were isolated from THP-1-derived macrophages ($5 \times 10^7$), and these samples were subsequently subjected to the ChIP assay kit (Sigma-Aldrich) following the manufacturer's instructions. Sheared DNA was incubated with primary antibodies against normal mouse IgG (as a negative control), TFAM (as a positive control), or the His-tagged protein (CdtB), along with magnetic protein A/G beads overnight at 4 °C on a rotor. The next day, immunoprecipitations were eluted, followed by reverse crosslinking and analysis via RT-qPCR utilizing 40 primer sets designed specifically for mitochondrial DNA. The primer sequences are listed in Supplementary Data 3.

## Determination of mitochondrial DNA copy number

Cells were harvested with DNA lysis buffer (Viagen) for genomic DNA extraction following the manufacturer's instructions. Genomic DNA (100 ng) was used for RT-qPCR analysis with primers complementary to the mitochondrial DNA sequence (mt-tRNA). The ΔCT value of mt-tRNA was normalized to the value of β-actin. The primer sequences are listed in Supplementary Data 3.

## Assessment of mitochondrial membrane potential

Mitochondrial membrane potential was measured using a JC-1 Assay Kit (Sigma) according to the protocol described. After staining with JC-1, fluorescence intensity was detected by a flow cytometer and analyzed by the ratio of JC-1 polymers.

## Mitochondria imaging by TMRE staining

Imaging of mitochondria was performed following established protocols. In brief, Henle-407 cells exposed to 160 ρM of typhoid toxin at 37 °C for 1 h were stained with the TMRE dye (Sigma) at 37 °C for 10 min in a humidified culture incubator maintained with 5% $CO_2$. The cells were washed with DPBS once and replenished with fresh medium. The cells stained with TMRE were then visualized using a Zeiss LSM780 confocal microscope. The resulting images were subjected to an analysis of mitochondrial morphology using Imaris software.

## T cell isolation and activation

Total T cells were isolated from the spleen and lymph nodes of 6–12-week-old C57BL/6 mice using anti-mouse Ig panning. The T cells were then cultured in the complete RPMI-1640 medium supplemented with 10% FBS. To activate the T cells, plate-bound anti-CD3 (4 µg/ml) plus anti-CD28 (2 µg/ml) were used in the presence of culture supernatant from mock-treated, WT typhoid toxin-treated, or the PltB$^{S35A}$ mutant of typhoid toxin. After five days of stimulation, T cells were collected and reactivated with phorbol ester 12-O-tetradec-anoylphorbol-13-acetate (TPA) (50 ng/ml) and PMA (500 ng/ml) for 5 h. The cells were then fixed using the Foxp3 staining kit buffer (eBioscience), and their live/dead cell status and expression levels of CD4, CD28, TIGIT, KLRG1, Foxp3, IFN-γ, and GzmB were determined using Northern Light spectrum Flow cytometry (Cytek Bioscience).

## SA-β-gal assay in T cells

The analyzed T cells were pelleted and resuspended in 500 µl of culture media containing 1.5 µl of Senescence dye (Abcam), followed by a 1-h incubation at 37 °C. After washing twice with Assay buffer XXVII (Abcam), the T cells were resuspended in 500 µl of Assay buffer XXVII and subjected to analysis using flow cytometry.

## T cell proliferation assay

Total mouse T cells were labeled with CellTrace Violet (Invitrogen) for proliferation tracking. Following labeling, T cells ($2 \times 10^5$) were activated using Dynabeads mouse T-activator CD3/CD28 beads (Invitrogen) and co-cultured with 20% of conditioned media collected from typhoid toxin-treated macrophages (20% v/v) for 6 days. After the incubation period, T cell proliferation was assessed by performing surface staining with anti-CD4 and CD8 antibodies, and subsequently analyzed using flow cytometry.

## Assessment of T cell cycle arrest

Mouse T cells were activated with Dynabeads Mouse T-Activator CD3/CD28 beads at a 2:1 bead-to-cell ratio and co-cultured with conditioned media collected from typhoid toxin-treated macrophages (20% v/v) for 3 and 6 days. After the co-culture period, T cells were surface-stained with anti-CD4 and CD8 antibodies, followed by fixation with 4% paraformaldehyde for 30 min. Subsequently, the fixed cells were treated with cold 70% ethanol at 4 °C. Following fixation, the cells were incubated in the permeabilization buffer (eBioscience) containing 0.25 µg of 7-AAD (BD Pharmingen) for 1 h at room temperature. After incubation, the cells were washed twice and analyzed using flow cytometry.

## Statistical analysis

The *P* values were calculated using a two-tailed, unpaired Student's *t*-test in GraphPad Prism (v.9). Differences were considered statistically significant when *P* values were below 0.05.

## Software

The following software was used in this study: CFX Manager™ Software (gene expression), ChemiDoc MP Imaging System and Image Lab (imaging and quantification of the band intensity of western blot), Imaris 9.7 (Oxford Instruments), Canvas X Draw & Adobe Photoshop (image preparation), FlowJo (analysis of flow cytometry data), CLC Genomics and MSigDB (RNA-seq analysis), BioRender.

## Reporting summary

Further information on research design is available in the Nature Portfolio Reporting Summary linked to this article.

## Data availability

The data discussed in this publication have been deposited in NCBI's Gene Expression Omnibus and are accessible through GEO Series accession number GSE247045. Source data are provided with this paper.

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

## Acknowledgements

We thank Drs. Zee-Fen Chang, Li-Chung Hsu, Jing-Jer Lin, and Shu-Chun Teng for discussion. We thank Dr. Chih-Ho Lai for providing the plasmids for CDT. We also thank the staff of the Biomedical Resource Core at the First Core Labs, College of Medicine, National Taiwan University, for technical assistance. This work was supported by the Taiwan National Science and Technology Council Grants MOST111-2636-B-002-028 (S.J.C.), NSTC112-2636-B-002-008 (S.J.C.), and a Taiwan Ministry of Education Grant 112V1404-4 (S.J.C.). Schematic illustration created with BioRender.

## Author contributions

H.Y.C. and S.J.C. conceived the project, and designed and interpreted experiments. H.Y.C. performed most of the experiments. W.C.H. carried out T cell functional experiments and interpreted experiments. H.Y.L. and P.Y.L. analyzed RNA-seq data. Y.C.L., Y.T.H., W.C.K., Y.J.H., and M.Y.L. performed additional experiments. S.C.H. and A.C.L. contributed to image analysis. G.G.L. contributed to the electron microscopy study. C.J.K. and H.C.T. supervised experiments and helped interpret data. S.J.C. directed the project and wrote the manuscript with comments from all authors.

## Competing interests

The authors declare no competing interests.
