## [Peer Review File · Nature Communications]

REVIEWER COMMENTS

Reviewer #1 (Remarks to the Author):

The manuscript entitled "Mitochondrial injury induced by a Salmonella genotoxin triggers the proinflammatory senescence-associated secretory phenotype" by Chen et al. has studied the effect of intoxication with the typhoid toxin on cellular senescence. The models used in the study are the human macrophage THP1 and epithelial Henle-407 cells, and murine primary activated CD4 positive T lymphocytes. Toxin containing a catalytically inactive CdtB subunit was used as negative control. Induction of senescence by TT in vitro and in vivo has been previously characterised (1-3), but the novelty of the manuscript resides in the description of mitochondrial damage as promoter of senescence and the associated-senescence secretory phenotype (SASP). However, there are many questions that were not addressed in the manuscript.

1. The authors do not mention at any stage which toxin concentration was used in the described experimental set up. This is relevant since an over-intoxication may produce results that are not physiologically relevant. Thus, a titration of the toxin effect should be tested. In addition, the authors do not specify the concentration of the inhibitors used (cGAS, p38 MAPK, JNK MAPK) in the Results, Material and Methods, Figure legends sections. This compromises the reproducibility of the experiments.
2. The effect of senescence was assayed at 16h post-intoxication. Considering that the cellular senescent is a late stage, which occurs upon failure of a proper DNA repair (in case of genotoxic-induced senescence, reviewed in 4), the authors should motivate the choice of the time point. This is relevant to distinguish between acute and sustained response to DNA damage, the latter conducive to the senescent phenotype.
3. Considering that the experimental set up was performed at 16h, it is not possible to exclude that damage of the genomic DNA would trigger part of the effect at early stages (acute response). Activation of the DNA damage response, type 1 interferon induction, and NFkB activation are very early effect (5-7). In addition, senescence in T cells induced by the cytolethal distending is partially dependent on the ATM kinase 2, a sensor of genomic DNA damage. It is relevant to understand at which time point, alteration of the mitochondrial activity and cytokine secretion is observed in relation to detection of the genomic DNA damage response, such as phosphorylation of the histone H2AX in the nucleus, which is observed at very early time points after intoxication with bacterial genotoxins. Is NFkB activated, if yes at which time point? Does ATM inhibitor prevent cytokine release and acquisition of the beta-galactosidase in typhoid toxin intoxicated cells at early or late effect?
In association with this point, does the cGAS KO cells present beta-galactosidase staining? It would be important to establish whether the senescent phenotype is uncoupled from the secretory phenotype in response to the typhoid toxin.
4. As a follow up of point 3 and in relation to figure 2b, the authors detected the presence of cytosolic dsDNA which can leak from the nucleus. In figure 3f, the authors showed that there was an increase in the mitochondrial DNA levels in the cytosol and a decreased mtDNA copy number. Can the authors

exclude the possibility that it is both mtDNA and nuclear DNA leaking to the cytosol? The authors showed that ROS production starts as 8h post-intoxication with WT typhoid toxin (figure 4b). This suggests that mitochondrial dysfunction is time-dependent. Have the authors checked if this correlates with expression of early markers for DNA damage and consequent “leakage” of DNA to the cytosol?.

5. It would be relevant to test whether the mitochondrial induced SASP is toxin specific effect, or it can be triggered by other DNA damaging agents such as ionizing radiation, compared to agents that would induce only cell cycle arrest as hydroxyurea, or double thymine block (time kinetics and titration experiments). If the effect is specific, it would indicate that the response is unique for the typhoid toxin and may be dependent on how the toxin traffics within the cell, accumulating preferentially in the mitochondria.

6. The authors stated that the secretory phenotype from intoxicated macrophages promotes senescence in CD4 positive T lymphocytes. However, the induction of cell cycle arrest and the beta-galactosidase staining was not performed in this cell type, thus the conclusion that the macrophage SASP induces senescence in T cells is not supported by the data presented.

7. The authors showed an enrichment of genes in the DNA sensing pathway upon intoxication with WT typhoid toxin in macrophages. Then the presence of cytosolic dsDNA is shown in Henle cells (according to the Material and Methods section, as this is not stated in the text nor in the figure legend). The authors previously showed (Suppl Figure 1) that the typhoid toxin induces senescence in several cell types, such as TH-P-1 macrophages and epithelial Henle cells. Have the authors checked that all the phenotypes they presented occurred in both cell types? So that the source of the DNA damage upon intoxication (= mitochondrial dysfunction) is the same in both cell types. The indistinct use of cell types without clearly stating which cell type the authors are using was done for several experiments. This lack of clarity prevents to determine how consistent are the results presented in the manuscript.

References

- 1 Ibler, A. E. M. et al. Typhoid toxin exhausts the RPA response to DNA replication stress driving senescence and Salmonella infection. *Nat Commun* 10, 4040, (2019).
- 2 Mathiasen, S. L. et al. Bacterial genotoxins induce T cell senescence. *Cell Reports* 35, (2021).
- 3 Martin, O. C. B. et al. Influence of the microenvironment on modulation of the host response by typhoid toxin. *Cell Rep* 35, 108931, (2021).
- 4 Gorgoulis, V. et al. Cellular Senescence: Defining a Path Forward. *Cell* 179, 813-827, (2019).
- 5 Lukas, C., Falck, J., Bartkova, J., Bartek, J. & Lukas, J. Distinct spatiotemporal dynamics of mammalian checkpoint regulators induced by DNA damage. *Nature Cell Biology* 5, 255-U212, (2003).
- 6 Wu, Z. H. et al. ATM- and NEMO-Dependent ELKS Ubiquitination Coordinates TAK1-Mediated IKK Activation in Response to Genotoxic Stress. *Molecular Cell* 40, 75-86, (2010).
- 7 Hartlova, A. et al. DNA damage primes the type I interferon system via the cytosolic DNA sensor STING to promote anti-microbial innate immunity. *Immunity* 42, 332-343, (2015).

Reviewer #2 (Remarks to the Author):

In the current study Han-Yi Chen et. al. describe the mechanism of by which Typhoid toxin from the human pathogen *Salmonella Typhi* induces senescence-associated secretory phenotype (SASP) in human macrophage-like cell line THP1. The study purported to show that Typhoid toxin triggers mitochondrial disruption that requires the DNase activity of the toxin leading to the SASP phenotype that is mediated by the release of mitochondrial DNA into the cytosol and activation of the cGAS/STING pathway. While these are interesting observations, the authors should address the following points to clarify some of the observations and strengthen the claims.

1. The major findings in this study relied on characterizations using THP-1 derived macrophages and should be confirmed using primary cells since many times cancer cell line observations don't recapitulate the biology of primary cells. Specifically, the authors should confirm cytokine differences and gene-expression differences of the most prominent genes in human monocyte-derived macrophages or murine bone marrow-derived macrophages. This will be a much more relevant model in which Typhoid toxin mediated differences can also be addressed in pro-inflammatory M1 vs anti-inflammatory M2 macrophages to see if SASP characteristics vary in these two very relevant macrophage populations.
2. In Fig panel 2b, why is dsDNA detected only with the antibody but not picked up by DAPI? To rule out any staining artifacts, some additional method of staining should be employed. It would also be important to distinguish the staining in Fig 2b as arising from mitochondrial DNA as opposed from micronuclei, nuclear membrane-encased chromosomal nuclei arising from DNA misrepair. For Fig. 2h p-STING bands were detected in STING KO cells. Why is that the case? Is the antibody non-specific?
3. In Figures 3 and 4, the authors describe mitochondrial damage induced by Typhoid toxin. In addition to mtDNA, mitochondrial damage could also lead to release of cytochrome c in the cytoplasm and intrinsic apoptosis pathway activation via Caspase 9. The authors should check if Typhoid toxin influences this pathway as well. Moreover, mtDNA does not only trigger type I interferon release via the cGAS/STING pathway but also could lead to the activation of other innate immune pathways, including the NLRP3 and AIM2 inflammasomes and the endosomal Toll-like receptor 9 (TLR9) signaling pathway. These have not been examined in this study.
4. The connection of senescence is not very strong in this study since for most phenotypes the authors just show inhibition of pro-inflammatory cytokine phenotype. This could also be just a mechanism of how Typhoid toxin functions in general and has nothing to do with senescence. To make a stronger case for the proposed senescence phenotype, B-gal staining (as shown in Fig. 1e) and additional senescence markers like phospho-p53, p21, p16 etc staining should be included for all the major experiments.
5. Finally, is the mechanism described in this study specific for Typhoid toxin or do other bacterial genotoxins, for example cytolethal distending toxins (CDTs) produced by gram-negative bacteria? Do they function similarly to induce mitochondrial damage leading to SASP? This investigation will advance the scope of the study and will be impactful in delineating the broader mechanism of action of bacterial genotoxins in general and not limit the study to only Typhoid toxin.

Reviewer #3 (Remarks to the Author):

In their submission, "Mitochondrial injury induced by a *Salmonella* genotoxin triggers the

proinflammatory senescence-associated secretory phenotype”, authors Chen et. al, provide evidence that during Salmonella Typhi infection, typhoid toxin promotes a senescent phenotype (SASP) through mitochondrial stress. Genotoxins impacting mitochondria is a novel and exciting concept and this work provides interesting observations, however more experiments uncovering the mechanism and better delineation of cellular consequences need to be performed.

Major concerns

1. Mechanism: It is unclear if the mitochondrial stress incurred in the presence of typhoid toxin is indirect and caused by damage to the nucleus or direct and caused by typhoid toxin interacting with the mitochondria. It is recommended that the authors uncouple this by using minimal systems, for example typhoid toxin WT and mutant and mitochondria incubated together. Authors should look at mtDNA damage, ROS, mitochondrial depolarization, and respiratory chain function. Does this also happen in ddC treated mitochondria or are they protected? During Salmonella infection inflammasome activation is also occurring, the authors propose that GSDMD might be forming pores in the mitochondria to promote the release of mtDNA. The authors should address this experimentally.

2. Readouts for SASP: SASP is a senescence-associated secretory phenotype (SASP), however the authors mainly rely on transcriptional readouts. This makes things a little muddy at times because certain modulations appear to have partial rescues. For example at times Type 1 IFN was reduced but no other components of the SASP? More clearly defining SASP and using cytokines released by the cells as SASP readouts is strongly recommended.

3. Cellular consequences: The authors show very cursory analysis of the impact of SASP on T cell function. While this is an interesting idea figure 6 needs to be heavily expanded on. What are the cellular consequences occurring in the macrophages driving this? Additionally, if mitochondrial stress and or mtDNA damage is driving the SASP would ddC treated or STING KO prevent T cell senescence? The authors should consider more T cell readouts and additional experiments involving some of the “rescues” they showed in earlier figures.

Minor concerns

In lines 155-156, the sentence “The cGAS-STING signaling pathway detects cytoplasmic DNA that may originate from damaged nuclear or mitochondrial DNA that is released from the stressed mitochondria.”, implies that nuclear DNA is being released from mitochondria, the sentence should be rewritten.

Line 262-263 “inactivation of STING inhibited the GCN2-mediated ISR 263 pathway”. The word Inactivation implies the pathway was activated then inhibited, I don’t believe that is the case since STING deficient THP-1 cells were used.

Reviewer #4 (Remarks to the Author):

Reviewer #5 (Remarks to the Author):

In this manuscript, Chen et al provides evidence on how Salmonella typhoid toxin targets mitochondria and triggers the SASP. As a microbial systems biologist, I have a few comments on the RNA-seq and pathway analysis done in the study.

1. Based on Fig 1a, it appears authors have performed triplicates for the transcriptomic analysis of mock, WT and PltB cells. For each replicate, can they show the RNA-seq reads such as tpm for each gene? These are missing from their source data.
2. Authors performed GSEA but they only showed pathways related to SASP (which favors their hypothesis) in Fig 1d. How do other pathways respond to typhoid toxin? Authors need to show ALL pathways from GSEA in the source data, comment on how SASP-related pathways are distributed among them, and whether other pathways are significantly enriched or depleted.
3. Minor: authors need to be more careful about showing the correct P-values. Many P-values in figures (eg, Fig 1c, Fig S1bcd) and source data (tab GSEA) are shown as 0. Please fix them.

REVIEWER COMMENTS

We thank the reviewers for their critical assessment of our manuscript and thoughtful feedback. Enclosed below is our comprehensive, point-by-point response to the comments provided by the five referees.

Reviewer #1 (Remarks to the Author):

The manuscript entitled "Mitochondrial injury induced by a Salmonella genotoxin triggers the proinflammatory senescence-associated secretory phenotype" by Chen et al. has studied the effect of intoxication with the typhoid toxin on cellular senescence. The models used in the study are the human macrophage THP1 and epithelial Henle-407 cells, and murine primary activated CD4 positive T lymphocytes. Toxin containing a catalytically inactive CdtB subunit was used as negative control. Induction of senescence by TT in vitro and in vivo has been previously characterised (1-3), but the novelty of the manuscript resides in the description of mitochondrial damage as promoter of senescence and the associated-senescence secretory phenotype (SASP). However, there are many questions that were not addressed in the manuscript.

1. The authors do not mention at any stage which toxin concentration was used in the described experimental set up. This is relevant since an over-intoxication may produce results that are not physiologically relevant. Thus, a titration of the toxin effect should be tested. In addition, the authors do not specify the concentration of the inhibitors used (cGAS, p38 MAPK, JNK MAPK) in the Results, Material and Methods, Figure legends sections. This compromises the reproducibility of the experiments.

We apologize for any previous lack of clarity and have made the suggested editorial revisions in the Methods section. The reviewer is correct; specifying the toxin concentration is a critical aspect of our methodology. In our experiments to assess gene expression and signaling pathways, we utilized a concentration of 160 μ M. Cells were treated for one hour and subsequently transitioned to regular culture medium. The choice of a one-hour incubation period was made to facilitate the synchronization of typhoid toxin entry into the cells. Furthermore, it is worth noting that this concentration falls within the typical titration range associated with physiologically relevant levels, as supported by previous research (PMID: 30951565, 31492859, 37792529). However, we understand the concern of potential over-intoxication, and we have performed a titration experiment to evaluate a range of toxin concentrations in our studies (see Figure 1 below). These additional findings demonstrate that the effect of typhoid toxin increases in a concentration-dependent manner. It is noteworthy that even after 16 hours of exposure to typhoid toxin, the cell morphology remained normal (see Figure 2 below). This evidence supports our contention that the concentration of typhoid toxin used in our study falls within a physiologically relevant range and accurately represents the biological responses to typhoid toxin without inducing non-physiological effects.

Regarding the concentration of inhibitors used, we have included this information in the revised manuscript to enhance the reproducibility of our work.

2. The effect of senescence was assayed at 16h post-intoxication. Considering that the cellular senescent is a late stage, which occurs upon failure of a proper DNA repair (in case of genotoxic-induced senescence, reviewed in 4), the authors should motivate the choice of the time point. This is relevant to distinguish between acute and sustained response to DNA damage, the latter conducive to the senescent phenotype.

We appreciate the Reviewer's comment and would like to clarify our rationale for the timing of our senescence assays. Cellular senescence is a complex biological process that unfolds over time, often as a result of failed DNA repair mechanisms. However, it is important to recognize that cellular senescence is not an abrupt event but rather a culmination of various molecular pathways and cellular stress responses that precede it. Typhoid toxin can trigger a rapid nuclear DNA damage response and cellular stress (PMID: 18191792, 31492859). These early events can subsequently lead to the activation of senescence-related signaling pathways. In our revised manuscript, we examined proinflammatory component expression at different time intervals. Our findings revealed that certain proinflammatory components could be induced at 8 hours post-treatment (shown in new Supplementary Fig.1a). This induction is, in part, attributable to the nuclear DNA damage caused by typhoid toxin, as demonstrated by the partial suppression of early inflammatory component expression with ATM inhibition (shown in new Supplementary Fig.2a). This effect was not as pronounced at the later time point (16 hours post-treatment) (shown in new Supplementary Fig.2b). Notably, the key cytokine IFN- β , known for its pivotal role in fortifying immune response and amplifying cellular senescence, was not induced at the early time point, but its induction became evident at 16 hours post-treatment (shown in new Supplementary Fig.1a). This observation underscores the dynamic

nature of typhoid toxin's action in terms of both timing and location. By examining the effect of senescence at the 16-hour time point, we can capture both the molecular responses and the ongoing processes leading to the development of the senescent phenotype. This approach provides us with valuable insights into the mechanisms through which typhoid toxin induces senescence and the associated signaling cascades, particularly by shedding light on mitochondrial damage that occurs at this later time point. It is important to note that this 16-hour time frame does not imply the full manifestation of senescence; instead, it serves as a snapshot of the events that contribute to this intricate process.

3. Considering that the experimental set up was performed at 16h, it is not possible to exclude that damage of the genomic DNA would trigger part of the effect at early stages (acute response). Activation of the DNA damage response, type 1 interferon induction, and NFkB activation are very early effect (5-7). In addition, senescence in T cells induced by the cytolethal distending is partially dependent on the ATM kinase 2, a sensor of genomic DNA damage. It is relevant to understand at which time point, alteration of the mitochondrial activity and cytokine secretion is observed in relation to detection of the genomic DNA damage response, such as phosphorylation of the histone H2AX in the nucleus, which is observed at very early time points after intoxication with bacterial genotoxins. Is NFkB activated, if yes at which time point? Does ATM inhibitor prevent cytokine release and acquisition of the beta-galactosidase in typhoid toxin intoxicated cells at early or late effect?

We appreciate the Reviewer's thorough assessment and questions regarding the timing of events in our study. Understanding the sequence of events is crucial, and we have made substantial efforts to address these concerns.

We acknowledge the potential contribution of genomic DNA damage to the acute response. We have now included a new figure in the revised manuscript, which demonstrates that early DNA responses to typhoid toxin can be detected as early as 8 hours post-treatment, characterized by the phosphorylation of H2AX (γ -H2AX) (shown in new Figure 1a). This emphasizes the acute effects of typhoid toxin on nuclear DNA. In our previous version, NF- κ B activation, noticeable as early as 8 hours post-typhoid toxin treatment, persists at a later time point of 16 hours (Figure 6c), with increased mRNA levels of proinflammatory components, such as IL-8, IL-1 β , and TNF- α (excluding IFN- β), at 8 hours (new Supplementary Fig.1a), suggesting the early activation of NF- κ B in response to typhoid toxin-induced nuclear DNA damage.

Given the relevance of ATM kinase in the phosphorylation of H2AX and its role in the nuclear DNA damage response, we tested the impact of ATM inhibition on this early response. Our findings revealed that ATM inhibition partially reduces mRNA levels of proinflammatory components at 8 hours post-treatment. However, it did not impede the production of these SASP factors after 16 hours of typhoid toxin exposure (new Supplementary Fig.2a,b). This aligns with the timing of nuclear DNA damage at the early stage, where ATM-mediated DNA damage response primarily contributes to the expression of proinflammatory components. Notably, IFN- β was only detectable at 16 hours following typhoid toxin treatment (new Supplementary Fig.1a). In our study, the role of mitochondrial damage and mtDNA is prominent in the late-stage senescence response, with mtDNA depletion experiments and mtROS overproduction (detected only at 16 hours and shown in new Figure 5b) reinforcing their significance. Moreover, ATM-deficient cells maintained a positive SA- β -gal phenotype at 48 hours following typhoid toxin exposure (new Supplementary Fig.2d,e), indicating the dispensable role of ATM in the development of SASP triggered by typhoid toxin. These

additional findings and clarifications comprehensively address the temporal aspects of our study and have been thoughtfully incorporated into the revised manuscript.

In association with this point, does the cGAS KO cells present beta-galactosidase staining? It would be important to establish whether the senescent phenotype is uncoupled from the secretory phenotype in response to the typhoid toxin.

We have indeed explored the link between the senescent phenotype and the secretory phenotype in the context of cGAS deficiency. In our revised manuscript, we have included results from experiments conducted with cGAS knockout (KO) cells, shedding light on the relationship between these phenotypes. The data reveal that cGAS KO cells still exhibited beta-galactosidase staining even 48 hours after exposure to the toxin (new Supplementary Fig. 8a-c), underlining their senescence phenotype. Interestingly, STING KO cells displayed a compromised senescence phenotype with reduced p16^{INK4a} expression (shown in new Supplementary Fig. 8d and Figure 2h). These observations imply distinct roles for cGAS and STING in the context of cellular senescence triggered by typhoid toxin. It appears that cGAS primarily initiates the production of proinflammatory SASP factors, while STING contributes to shaping the senescence phenotype and collaborates with cGAS in orchestrating the SASP response. This new data significantly enriches our understanding of the complex dynamics involved in these cellular responses and their regulation.

4. As a follow up of point 3 and in relation to figure 2b, the authors detected the presence of cytosolic dsDNA which can leak from the nucleus. In figure 3f, the authors showed that there was an increase in the mitochondrial DNA levels in the cytosol and a decreased mtDNA copy number. Can the authors exclude the possibility that it is both mtDNA and nuclear DNA leaking to the cytosol?

In our previous manuscript, we did not definitively exclude the possibility that cytosolic dsDNA could originate from both mtDNA and nuclear DNA. Our findings provide substantial evidence of mtDNA leakage into the cytosol and its pivotal role in activating the STING signaling pathway, as supported by our RT-qPCR analysis of mtDNA-specific sequences in the cytosolic fraction of cells exposed to the toxin. We appreciate the Reviewer's valuable observations concerning the sources of cytosolic DNA, and we have taken steps to address this concern more comprehensively. We expanded our analysis by employing immunofluorescence microscopy. In doing so, we incorporated a nuclear DNA damage marker, γ -H2AX, and a mitochondrial marker, COX4, to distinguish the origin of cytosolic DNA, whether from the nucleus or mitochondria. The results revealed the presence of nuclear DNA (cytosolic dsDNA with γ -H2AX staining) in the cytosol following typhoid toxin treatment, while also detecting cytosolic DNA without γ -H2AX signal (See Figure 3 below). Furthermore, we applied the mitochondrial transcription factor TFAM as a marker, co-stained with a fluorescent DNA probe (PicoGreen) and TOMM20 in cells exposed to typhoid toxin. This showed that some cytosolic DNA colocalized with TFAM (shown in new Figure 3f,g), providing additional support for the mitochondrial origin of cytosolic DNA. Taken together, our results demonstrate the presence of both mitochondrial DNA and nuclear DNA in the cytosol following exposure to typhoid toxin. Importantly, it is the presence of mitochondrial DNA outside the mitochondria that plays a pivotal role in initiating the STING-dependent SASP response.

The authors showed that ROS production starts as 8h post-intoxication with WT typhoid toxin (figure 4b). This suggests that mitochondrial dysfunction is time-dependent. Have the authors checked if this correlates with expression of early markers for DNA damage and consequent “leakage” of DNA to the cytosol?.

Indeed, we have investigated these aspects, and our data revealed that the nuclear DNA damage response, as evidenced by γ -H2AX staining, and total ROS production commence at 8 hours post-intoxication with typhoid toxin, as illustrated in new Figure 1a and Figure 5a. This order of events suggests that the nuclear DNA damage response occurs prior to the onset of mtROS overproduction, which is only observed at 16 hours post-treatment, as depicted in our new Figure 5b within the manuscript. Given the time-dependent nature of these observations, we further explored the presence of cytosolic mtDNA and its potential connection to nuclear DNA damage response. However, we do not believe that the mtDNA leakage into the cytosol is a consequence of the nuclear DNA damage response, and here is why:

First, mitochondrial reactive oxygen species (mtROS) play a key role in mtDNA leakage induced by typhoid toxin. Notably, the total ROS production became evident as early as 8 hours after treatment (shown in new Figure 5a), coinciding with the initiation of the nuclear DNA damage response. However, the event of mtDNA leakage occurs only at 16 hours (shown in new Figure 3h, i). This indicates that ROS generated from the nuclear DNA damage response alone is insufficient to trigger the release of mtDNA into the cytosol. It is essential to underscore that the release of mtDNA into the cytosol is a distinct and delayed event that becomes evident only at the 16-hour time point.

Second, we conducted experiments inhibiting the kinase (ATM) responsible for phosphorylating H2AX but found that it did not effectively suppress the SASP factors after 16 hours of typhoid toxin exposure (new Supplementary Fig. 2). This suggests that the signaling from nuclear DNA damage mediated by the ATM kinase is not involved in typhoid toxin-induced SASP through mitochondrial damage.

Third, our new results depicted in Figure 4 b,c demonstrate that typhoid toxin directly binds to mtDNA, leading to mtDNA damage. These findings strongly support the idea that mitochondrial damage is the primary driver behind mtDNA leakage into the cytosol.

Collectively, our data highlight the crucial role of mitochondrial damage induced by typhoid toxin in the regulation of the SASP response. The direct involvement of the nuclear DNA

damage response remains a topic for further investigation. We have integrated these insights and explanations into our revised manuscript.

5. It would be relevant to test whether the mitochondrial induced SASP is toxin specific effect, or it can be triggered by other DNA damaging agents such as ionizing radiation, compared to agents that would induce only cell cycle arrest as hydroxyurea, or double thymine block (time kinetics and titration experiments). If the effect is specific, it would indicate that the response is unique for the typhoid toxin and may be dependent on how the toxin traffics within the cell, accumulating preferentially in the mitochondria.

We thank the Reviewer's valuable suggestion. We have conducted a series of experiments, which included examining mtDNA leakage, assessing mtDNA copy number, and evaluating the expression of SASP factors (shown in new Supplementary Fig. 22). These experiments were designed to compare the effects of typhoid toxin with those induced by other DNA-damaging agents, such as hydroxyurea. In our revised manuscript, we present the data demonstrating that even when cells were exposed to higher concentrations of hydroxyurea, we did not observe the release of mtDNA into the cytosol or a reduction in mtDNA copy number at both early and late stages after treatment. Additionally, there was no significant increase in mRNA levels of proinflammatory components. These results emphasize the specificity of the SASP response triggered by typhoid toxin. The Reviewer is correct. These findings further suggest that typhoid toxin's intracellular trafficking may play a pivotal role, directing its entry into the mitochondria and influencing the observed effects. We have thoughtfully incorporated these findings into our revised manuscript to highlight the significance of typhoid toxin's unique mode of action.

6. The authors stated that the secretory phenotype from intoxicated macrophages promotes senescence in CD4 positive T lymphocytes. However, the induction of cell cycle arrest and the beta-galactosidase staining was not performed in this cell type, thus the conclusion that the macrophage SASP induces senescence in T cells is not supported by the data presented.

We appreciate the Reviewer's insightful suggestion, and we have conducted additional experiments to directly assess the senescence effect in T cells. We have now included these results in the revised manuscript (shown in new Figure 7 and Supplementary Fig. 19). Our new data demonstrates that exposure of the conditioned medium obtained from WT toxin-treated macrophages to CD4 and CD8 T cells resulted in a significant increase in the cell population in the G₂/M phase (new Supplementary Fig. 19 a,d) and SA- β -Gal activity (new Supplementary Fig. 19 c,f). Furthermore, we observed a reduction in cell proliferation in both T cells exposed to this conditioned medium (in new Figure 7a and Supplementary Fig. 19). These results provide strong evidence of a senescent bystander effect on T cells induced by typhoid toxin.

7. The authors showed an enrichment of genes in the DNA sensing pathway upon intoxication with WT typhoid toxin in macrophages. Then the presence of cytosolic dsDNA is shown in Henle cells (according to the Material and Methods section, as this is not stated in the text nor in the figure legend). The authors previously showed (Suppl Figure 1) that the typhoid toxin induces senescence in several cell types, such as THP-1 macrophages and epithelial Henle cells. Have the authors checked that all the phenotypes they presented occurred in both cell types? So that the source of the DNA damage upon intoxication (= mitochondrial dysfunction) is the same in both cell types. The indistinct use of cell types without clearly stating which cell type the authors are using was done for several experiments. This lack of clarity prevents to determine how consistent are the results presented in the manuscript.

We apologize for this confusion and appreciate the Reviewer's comment regarding the need for specifying the cell types used in our study. We have addressed this concern and made the necessary editorial changes in the revised manuscript. We demonstrate that typhoid toxin can induce cellular senescence and the SASP in Henle-407 cells (in new Supplementary Fig. 4). Additionally, we confirmed that the phenotypes indicative of mitochondrial damage, such as the presence of mtDNA in the cytosol and a reduction in mtDNA copy number, were consistently observed in both THP-1-derived macrophages and Henle-407 intestinal epithelial cells (shown in new Figures 3h, i and 4d). These findings collectively highlight the robustness and consistency of the typhoid toxin-induced SASP resulting from mitochondrial damage.

References

- 1 Ibler, A. E. M. et al. Typhoid toxin exhausts the RPA response to DNA replication stress driving senescence and Salmonella infection. *Nat Commun* 10, 4040, (2019).
- 2 Mathiasen, S. L. et al. Bacterial genotoxins induce T cell senescence. *Cell Reports* 35, (2021).
- 3 Martin, O. C. B. et al. Influence of the microenvironment on modulation of the host response by typhoid toxin. *Cell Rep* 35, 108931, (2021).
- 4 Gorgoulis, V. et al. Cellular Senescence: Defining a Path Forward. *Cell* 179, 813-827, (2019).
- 5 Lukas, C., Falck, J., Bartkova, J., Bartek, J. & Lukas, J. Distinct spatiotemporal dynamics of mammalian checkpoint regulators induced by DNA damage. *Nature Cell Biology* 5, 255-U212, (2003).
- 6 Wu, Z. H. et al. ATM- and NEMO-Dependent ELKS Ubiquitination Coordinates TAK1-Mediated IKK Activation in Response to Genotoxic Stress. *Molecular Cell* 40, 75-86, (2010).
- 7 Hartlova, A. et al. DNA damage primes the type I interferon system via the cytosolic DNA sensor STING to promote anti-microbial innate immunity. *Immunity* 42, 332-343, (2015).

Reviewer #2 (Remarks to the Author):

In the current study Han-Yi Chen et. al. describe the mechanism of by which Typhoid toxin from the human pathogen Salmonella Typhi induces senescence-associated secretory phenotype (SASP) in human macrophage-like cell line THP1. The study purported to show that Typhoid toxin triggers mitochondrial disruption that requires the DNase activity of the toxin leading to the SASP phenotype that is mediated by the release of mitochondrial DNA into the cytosol and activation of the cGAS/STING pathway. While these are interesting observations, the authors should address the following points to clarify some of the observations and strengthen the claims.

1. The major findings in this study relied on characterizations using THP-1 derived macrophages and should be confirmed using primary cells since many times cancer cell line observations don't recapitulate the biology of primary cells. Specifically, the authors should confirm cytokine differences and gene-expression differences of the most prominent genes in human monocyte-derived macrophages or murine bone marrow-derived macrophages. This will be a much more relevant model in which Typhoid toxin mediated differences can also be addressed in pro-inflammatory M1 vs anti-inflammatory M2 macrophages to see if SASP characteristics vary in these two very relevant macrophage populations.

We acknowledge the importance of validating our results in primary cells, as suggested by the Reviewer. To confirm our findings, we assessed the gene expression of proinflammatory

components in murine bone marrow-derived macrophages (BMDMs) exposed to typhoid toxin. Consistent with our observations in THP-1-derived macrophages, typhoid toxin exposure in BMDMs significantly increased the mRNA levels of proinflammatory factors, including *Ifnb*, *Cxcl2* (murine CXCL2 is functional homolog of human IL-8), *Il1b*, *Il6*, *Il18*, and *Tnf* (shown in new Figure 1j and Figure 2f). These results conclusively demonstrate that the proinflammatory SASP occurs in BMDMs, further emphasizing the impact of typhoid toxin under physiological conditions.

In response to the Reviewer's suggestion, we explored potential differences in SASP characteristics between pro-inflammatory M1 and anti-inflammatory M2 macrophages (new Supplementary Fig. 3 and Fig. 5), providing a comprehensive understanding of typhoid toxin's influence on diverse macrophage subsets. We found that both primary M1 and M2 macrophages produced *Ifnb*, *Cxcl2*, *Il6*, and *Tnf*. However, notable distinctions in gene expression patterns emerged between M1 and M2 macrophages. Specifically, M1 macrophages displayed significantly elevated mRNA levels of *Il1b* and *Il10*, while M2 macrophages notably expressed *Il18*. These observations suggest that typhoid toxin can exert its effects on both macrophage types through a common molecular mechanism while influencing specific gene expression patterns in M1 and M2 macrophages. These variations may be mediated by unique cellular signaling pathways and epigenetic modulations. Collectively, these results deepen our comprehension of the broader impact of typhoid toxin on different macrophage populations.

2. In Fig panel 2b, why is dsDNA detected only with the antibody but not picked up by DAPI? To rule out any staining artifacts, some additional method of staining should be employed. It would also be important to distinguish the staining in Fig 2b as arising from mitochondrial DNA as opposed from micronuclei, nuclear membrane-encased chromosomal nuclei arising from DNA misrepair.

We appreciate the Reviewer's attention and apologize for this confusion. In the previous version of the manuscript, we observed dsDNA detection with the antibody and DAPI staining when enhancing the fluorescent DAPI signal. To mitigate this confusion, we have replaced an image that exhibits a more pronounced DAPI signal (shown in new Figure 2b). To address the Reviewer's concerns and further clarify our findings, we conducted additional experiments to distinguish between mitochondrial DNA and micronuclei. In response to Reviewers 1 and 4, we utilized mitochondrial transcription factor TFAM as a marker, in conjunction with co-staining using a fluorescent DNA probe (PicoGreen) and TOMM20 in cells exposed to typhoid toxin. This analysis revealed co-localization of cytosolic DNA with TFAM outside the mitochondria, as depicted in the new Figure 3f, g, further substantiating the mitochondrial origin of cytosolic DNA. These additional findings further support the presence of cytosolic mtDNA and strengthen the quality and integrity of our results, addressing the Reviewer's concerns effectively.

For Fig. 2h p-STING bands were detected in STING KO cells. Why is that the case? Is the antibody non-specific?

We appreciate the Reviewer's attention to this matter and would like to clarify the observation in Figure 2h, as presented in the previous version of the manuscript. We are aware that the presence of non-specific bands in western blot experiments can occasionally be a challenge, possibly due to the high antibody concentration. In our study, we utilized the antibody targeting p-STING (the rabbit mAb #50907 obtained from Cell Signaling), which has been employed in numerous investigations. However, we recognize that non-specific binding can, on occasion,

yield misleading results. To address this concern, we conducted additional experiments and updated the results in the revised manuscript by titrating the antibody in western blot assays. Our new data provide strong evidence of the antibody's specificity, as p-STING bands are not detected in STING KO cells. We apologize for any confusion arising from the previous presentation of the data and appreciate the opportunity to rectify this issue in the updated manuscript.

3. In Figures 3 and 4, the authors describe mitochondrial damage induced by Typhoid toxin. In addition to mtDNA, mitochondrial damage could also lead to release of cytochrome c in the cytoplasm and intrinsic apoptosis pathway activation via Caspase 9. The authors should check if Typhoid toxin influences this pathway as well. Moreover, mtDNA does not only trigger type I interferon release via the cGAS/STING pathway but also could lead to the activation of other innate immune pathways, including the NLRP3 and AIM2 inflammasomes and the endosomal Toll-like receptor 9 (TLR9) signaling pathway. These have not been examined in this study.

We thank the Reviewer for the suggestion. To investigate caspase-9 activation in typhoid toxin-treated cells, we conducted western blot analysis using an anti-caspase-9 antibody (Cat. 9508 obtained from Cell Signaling). Our results revealed the presence of a processed product, indicative of the active form of caspase-9, in THP-1-derived macrophages following treatment with WT typhoid toxin (shown in new Supplementary Fig.15a). This active form of caspase-9 was conspicuously absent in cells treated with the PltB mutant of typhoid toxin. These findings suggest that mitochondrial damage induced by typhoid toxin can lead to the release of cytochrome c, thereby initiating the activation of caspase-9.

Following the Reviewer's suggestion, we conducted additional experiments to investigate NLRP3 and AIM inflammasome activation. Our results demonstrate that treatment with WT typhoid toxin in THP-1-derived macrophages significantly upregulated the expression of NLRP3 and the activation of gasdermin D (GSDMD) (in new Supplementary Fig.15a), as evidenced by western blot analysis. These findings provide compelling evidence for the activation of the NLRP3 inflammasome in cells treated with typhoid toxin. To investigate the potential activation of the AIM2 inflammasome, we employed immunofluorescence microscopy to monitor endogenous AIM2 assembly, characterized by puncta formation within cells. As the results showed, neither the mutant nor the WT typhoid toxin induced AIM2 inflammasome activation (in new Supplementary Fig.15b,c). AIM2 oligomerization was only observed under the irradiation condition, indicating that typhoid toxin cannot induce AIM2 inflammasome activation.

Regarding the activation of TLR9, a crucial immune sensor primarily located within endosomal-lysosomal compartments, our proposed model makes it unlikely for TLR9 to be activated through cytosolic mtDNA. Furthermore, there is a practical limitation in investigating TLR9 activation in THP-1 cells, given the absence of TLR9 expression in this cell line. To address this, we examined TLR9 activation in murine RAW264.7 macrophages and primary bone marrow-derived macrophages (BMDMs). Our analyses revealed that typhoid toxin does not induce an increase in TLR9, except in the case of a positive control (see Figure 4 below), CpG. These results suggest that typhoid toxin's impact on TLR9 activation is negligible, thus providing clarity regarding the specific immune pathways impacted by the toxin.

Taken together, the information expands our understanding of the specific pathways influenced by typhoid toxin and provides valuable insights into the host immune response to this toxin.

4. The connection of senescence is not very strong in this study since for most phenotypes the authors just show inhibition of pro-inflammatory cytokine phenotype. This could also be just a mechanism of how Typhoid toxin functions in general and has nothing to do with senescence. To make a stronger case for the proposed senescence phenotype, B-gal staining (as shown in Fig. 1e) and additional senescence markers like phospho-p53, p21, p16 etc staining should be included for all the major experiments.

We value the insightful feedback from the Reviewer regarding the connection between senescence and the observed phenotypes in our study. The phenomenon of typhoid toxin-induced senescence has been reported by Humphreys' laboratory (PMID: 31492859, 37792529), and in our investigation, we have presented evidence of this senescence phenotype in both macrophages and intestinal epithelial cells, primarily by detecting SA- β -gal activity (shown in Figure 1b,c and new Supplementary Fig. 4a,b). To strengthen the association between the responses elicited by typhoid toxin and senescence, we have integrated additional senescence markers into our primary experiments. Specifically, we validated the phosphorylation of p53 and the induction of p16^{INK4a} in typhoid toxin-treated cells, including THP-1-derived macrophages and Henle-407 intestinal epithelial cells. These results are now included in the revised manuscript (shown in new Figure 1a and Supplementary Fig. 4d). To explore the link between typhoid toxin-induced mtROS overproduction and cellular senescence, we performed experiments to investigate whether blocking mtROS affected senescence development in typhoid toxin-treated cells. The results demonstrated that the addition of MitoQ significantly led to a significant reduction in the number of SA- β -gal-positive senescent cells in response to typhoid toxin (see new Supplementary Fig. 14), accompanied by a notable decrease in the induction of p16^{INK4a} (shown in new Figure 5c). It is important to note that the phosphorylation of p53 remained unaffected by the administration of MitoQ (see new Figure 5c). Given that p53 activation plays a pivotal role in cellular senescence in response to nuclear DNA damage, we speculate that its activation might not be influenced by mtROS neutralization. Furthermore, the significant role of STING in cellular senescence and the SASP has been reported (PMID: 28759028). In our study, we demonstrated that STING KO cells failed to induce p16^{INK4a} expression (shown in new Figure 2h). These new findings with the additional markers provide substantial support for the concept that typhoid toxin-induced senescence is intricately linked to the SASP through mitochondrial damage.

5. Finally, is the mechanism described in this study specific for Typhoid toxin or do other bacterial genotoxins, for example cytolethal distending toxins (CDTs) produced by gram-negative bacteria? Do they function similarly to induce mitochondrial damage leading to SASP? This investigation will advance the scope of the study and will be impactful in delineating the broader mechanism of action of bacterial genotoxins in general and not limit the study to only typhoid toxin.

We thank the Reviewer for this excellent suggestion. To investigate whether CDT has a similar effect in inducing the SASP by targeting mitochondria, we conducted several assays. First, we assessed the production of proinflammatory components in CDT-treated cells and observed an increase in several proinflammatory factors, except for IFN- β (shown in new Figure 4e). This suggests a distinct mechanism for CDT-induced cellular senescence. Importantly, one key distinction arises from the fact that CDT toxin cannot access the mitochondria (see new Supplementary Fig. 11b). Consequently, we did not detect the release of mtDNA into the cytosol and observed no reduction in mtDNA copy number in cells exposed to CDT (shown in new Figure 4f,g). It has been revealed that CDT triggers senescence through the ATM-p38 axis in T cells (PMID: 34107253), and we demonstrated that ATM is not required in typhoid toxin-induced senescence and the SASP in macrophages. Hence, our findings, in conjunction with recent studies, collectively indicate that the SASP induced by CDT is not dependent on mitochondrial damage. This divergence in mechanism might be attributed to how typhoid toxin specifically interacts with and accesses the mitochondria, setting it apart from CDT.

Reviewer #3 (Remarks to the Author):

In their submission, "Mitochondrial injury induced by a Salmonella genotoxin triggers the proinflammatory senescence-associated secretory phenotype", authors Chen et. al, provide evidence that during Salmonella Typhi infection, typhoid toxin promotes a senescent phenotype (SASP) through mitochondrial stress. Genotoxins impacting mitochondria is a novel and exciting concept and this work provides interesting observations, however more experiments uncovering the mechanism and better delineation of cellular consequences need to be performed.

Major concerns

1. Mechanism: It is unclear if the mitochondrial stress incurred in the presence of typhoid toxin is indirect and caused by damage to the nucleus or direct and caused by typhoid toxin interacting with the mitochondria. It is recommended that the authors uncouple this by using minimal systems, for example typhoid toxin WT and mutant and mitochondria incubated together. Authors should look at mtDNA damage, ROS, mitochondrial depolarization, and respiratory chain function. Does this also happen in ddC treated mitochondria or are they protected?

We appreciate the Reviewer's suggestion concerning the mechanism of mitochondrial stress induced by typhoid toxin. While we concur that the involvement of damaged nuclear DNA in cellular senescence induced by typhoid toxin cannot be ruled out, there are inherent limitations when dealing with a minimal system, primarily due to the challenge of typhoid toxin entry. Typhoid toxin alone cannot enter isolated mitochondria (see Figure 5 below), making it difficult to assess its interaction with mitochondria independently. We believe that typhoid toxin's access to the mitochondria may necessitate membrane vesicle transport within the cells.

However, in response to the Reviewer's concerns, we conducted a mitochondrial genome-wide ChIP assay (mtDIP) to explore whether typhoid toxin directly targets mitochondrial DNA (shown in new Figure 4b). This assay allowed us to investigate the toxin's direct impact on mitochondrial DNA, including its binding affinity and potential damage to mitochondrial DNA. Our results demonstrated that typhoid toxin can bind to mitochondrial DNA, with a notable affinity for specific regions, such as the D-loop, leading to mitochondrial DNA damage. These findings provide important insights into the direct interaction of typhoid toxin with mitochondrial DNA and its role in initiating mitochondrial damage, which in turn triggers the proinflammatory SASP. This new evidence enhances our understanding of the mechanisms underpinning typhoid toxin-induced mitochondrial stress and its downstream consequences.

During Salmonella infection inflammasome activation is also occurring, the authors propose that GSDMD might be forming pores in the mitochondria to promote the release of mtDNA. The authors should address this experimentally.

We proposed that GSDMD oligomerization in the mitochondria, leading to pore formation, could serve as a mechanism for releasing mtDNA into the cytosol. However, we found that inhibiting GSDMD did not prevent the leakage of mtDNA into the cytosol or reduce the production of SASP components (see new Supplementary Fig. 15d-f). Therefore, these results suggest that GSDMD activation may not be the primary mechanism responsible for the release of mtDNA into the cytosol in typhoid toxin-treated cells.

2. Readouts for SASP: SASP is a senescence-associated secretory phenotype (SASP), however the authors mainly rely on transcriptional readouts. This makes things a little muddy at times because certain modulations appear to have partial rescues. For example at times Type 1 IFN was reduced but no other components of the SASP? More clearly defining SASP and using cytokines released by the cells as SASP readouts is strongly recommended.

We thank the Reviewer's suggestion regarding the assessment of the SASP and the need for more readouts. To address this concern, we have made significant improvements in our revised manuscript. We conducted a cytokine dot blot array to examine the SASP (shown in new Figure 1i). This approach enabled us to detect various cytokines, specifically in cells exposed to WT typhoid toxin, but not the PltB mutant. These identified cytokines are well-established indicators of the SASP. Furthermore, we have included data that validates the protein levels of IFN- β , TNF- α , and IL-8 in WT typhoid toxin-treated cells (see new Figure 2e, Supplementary Fig. 1e and Fig. 6). Collectively, these findings provide robust evidence supporting the presence of the SASP in cells in response to typhoid toxin.

3. Cellular consequences: The authors show very cursory analysis of the impact of SASP on T cell function. While this is an interesting idea figure 6 needs to be heavily expanded on. What are the cellular consequences occurring in the macrophages driving this? Additionally, if mitochondrial stress and or mtDNA damage is driving the SASP would ddC treated or STING

KO prevent T cell senescence? The authors should consider more T cell readouts and additional experiments involving some of the “rescues” they showed in earlier figures.

Paracrine senescence (senescent bystander effect) plays a pivotal role in the tissue environment. In our study, we provide evidence of senescence and the SASP occurring in macrophages and intestinal epithelial cells. These cells release proinflammatory SASP factors, which are known to attract immune cells, and we demonstrated that these factors can induce senescence in neighboring cells, including T cells. Proper immune surveillance is crucial for controlling *Salmonella* infection. However, when immune senescence becomes dysregulated, it can lead to a decline in immunity and the persistence of a detrimental SASP, potentially resulting in chronic inflammation and severe tissue pathology. While we could not conduct in vivo experiments due to the exclusive human host nature of *Salmonella* Typhi, it is common to observe severe tissue damage in typhoid fever patients who do not receive therapeutic interventions. We have addressed the Reviewer’s suggestion by including more T cell readouts in our revised manuscript (shown in new Figure 7 and Supplementary Fig.19-21). We present additional assays that confirm the senescence phenotype in T cells triggered by senescent macrophages. These assays assessed the proliferation of activated CD4 and CD8 T cells exposed to conditioned medium from typhoid toxin-treated cells. The results show reduced proliferation, cell death, and arrested cell cycle in T cells, aligning with other marker expressions presented in the previous version of the manuscript. These findings shed further light on the consequences of the SASP on T cell function. Significantly, when we depleted the “fire” (mitochondrial DNA) that ignites the STING signaling pathway in macrophages or used STING-deficient cells, we observed a remarkable reduction in paracrine senescence in T cells. This underscores the critical role of mitochondrial DNA in initiating the SASP in cells exposed to typhoid toxin. Collectively, our findings emphasize that typhoid toxin not only impairs its target cells but also influences on neighboring cell types during intoxication through a senescent bystander effect. These insights are now incorporated into the updated manuscript to provide a more comprehensive understanding of the broader consequences of the SASP.

Minor concerns

In lines 155-156, the sentence “The cGAS-STING signaling pathway detects cytoplasmic DNA that may originate from damaged nuclear or mitochondrial DNA that is released from the stressed mitochondria.”, implies that nuclear DNA is being released from mitochondria, the sentence should be rewritten.

Thank you for pointing out the potential confusion in our sentence. We have corrected this in the revised manuscript.

Line 262-263 “inactivation of STING inhibited the GCN2-mediated ISR 263 pathway”. The word Inactivation implies the pathway was activated then inhibited, I don’t believe that is the case since STING deficient THP-1 cells were used.

We thank the Reviewer for pointing this out, and we agree that the term “inactivation” might not accurately convey the situation in this context. We have updated the sentence to better reflect the experimental setup and results.

Reviewer #4 (Remarks to the Author):

We appreciate the valuable feedback and suggestions provided by the Reviewer. In response, we have made necessary editorial changes and performed additional experiments to address the concerns in the revised manuscript comprehensively.

Reviewer #5 (Remarks to the Author):

In this manuscript, Chen et al provides evidence on how Salmonella typhoid toxin targets mitochondria and triggers the SASP. As a microbial systems biologist, I have a few comments on the RNA-seq and pathway analysis done in the study.

1. Based on Fig 1a, it appears authors have performed triplicates for the transcriptomic analysis of mock, WT and PtlB cells. For each replicate, can they show the RNA-seq reads such as tpm for each gene? These are missing from their source data.

Thank you for the Reviewer's suggestion regarding our RNA-seq data. We have included the transcript per million (TPM) values for reference.

2. Authors performed GSEA but they only showed pathways related to SASP (which favors their hypothesis) in Fig 1d. How do other pathways respond to typhoid toxin? Authors need to show ALL pathways from GSEA in the source data, comment on how SASP-related pathways are distributed among them, and whether other pathways are significantly enriched or depleted.

We value the Reviewer's suggestion, and we recognize the significance of offering a more thorough insight into the pathways impacted by typhoid toxin. In the revised manuscript, we have included all KEGG pathway enrichment results from RNA-seq data with a significance threshold of P -value < 0.05 and q -value < 0.25 in the source data. Furthermore, we have expanded our discussion in the manuscript, addressing the impact of these SASP-related pathways in the Discussion section. This improvement further clarifies the connections between these pathways, senescence, and the SASP.

3. Minor: authors need to be more careful about showing the correct P-values. Many P-values in figures (eg, Fig 1c, Fig S1bcd) and source data (tab GSEA) are shown as 0. Please fix them.

We apologize for this confusion and appreciate the Reviewer pointing out this. We have made these changes in the figures and supplementary data.

REVIEWERS' COMMENTS

Reviewer #3 (Remarks to the Author):

The reviewers have done a good job addressing my critiques and concerns. Good job!

Reviewer #5 (Remarks to the Author):

Authors have satisfyingly addressed all my previous comments.

Reviewer #7 (Remarks to the Author):

The manuscript entitled "Mitochondrial injury induced by a Salmonella genotoxin triggers the proinflammatory senescence-associated secretory phenotype" by Chen et al. has studied the effect of intoxication with the typhoid toxin on cellular senescence. The author has revised the paper according to the opinions of the reviewers, but there are still some minor problems that need to be corrected.

1. Figure 2h, Why is there no difference in cGAS between WT and PltbS35A after STING KO?
2. The last three plots of Figure 6d are not labeled with detection index names.
3. Please provide verification of the effect of mitoQ on scavenging ROS in macrophages.

Reviewer #8 (Remarks to the Author):

A very interesting manuscript supported by high-quality data. Typhoid toxin penetration of mitochondria and the leakage of mitochondrial DNA into the cytosol leading to a SASP-like response in macrophages is interesting and novel.

The authors have done a good job addressing reviewer comments. My only concern relates to point (iv) below. The authors state the macrophages are senescent but there is so much overlap with immunostimulated macrophages that it is difficult to determine whether the macrophages are senescent or undergoing some form of novel immunostimulation due to the toxin entering the mitochondria. The markers p16Ink4a/SA β G they use are at short timepoints (e.g. 16h) and are known to be reversed in macrophages after several days, which indicates they may in fact not be senescent, more senescent-like. This does not detract from the many interesting findings in the manuscript but I think the manuscript should be clear about this limitation in the discussion.

I have only minor suggestions to make the text more clear:

(i) line 30 in the abstract. 'essential genotoxin of Salmonella Typhi'. It is not clear what 'essential' refers to. Toxin-negative S.Typhi still cause typhoid fever in human participants (Gibani et al 2019) so the word 'essential' is misleading.

(ii) Following on from this, line 49-50 in the introduction: ‘...ethical considerations limit its in-depth investigation in human volunteer studies’. Human infection challenge studies have been reported (Gibani et al 2019). I think the reference should be included and/or sentence rephrased. [SEP]

(iii) Line 116-119 - citation of Fig 1F relating to GSEA is missing

(iv) I tend to agree with reviewer 2, point 4 regarding the determination of senescence in the macrophages, which is still not clearly addressed experimentally and I suggest modest changes to the discussion to reflect this limitation. See below.

There is a lack of background or discussion on macrophage senescence, which appear similar to immunostimulated macrophages. Are the macrophage cells senescent? A defining feature of senescent cells is their durable cell cycle arrest in response to cell stress but this metric cannot be applied to the polarized THP1 macrophage. The authors observe senescence markers p-p53, γ H2AX, and notably elevated p16Ink4a and SA β G. However, p16Ink4a/SA β G-signalling has been observed in macrophages as part of their natural physiological response to inflammatory stimuli and is reversed days after (<https://doi.org/10.18632/aging.101268>). Moreover, stimulated macrophages have an inflammatory secretome, which overlaps with the identified SASP factors, eg. IL6, IL1b, so it is perhaps not surprising a ‘SASP-like’ phenotype is enriched. There is significant debate regarding macrophage senescence and how it is defined (<https://doi.org/10.1083/jcb.202010162>). The manuscript contributes to this debate with their findings and this should be acknowledged and made clear within their discussion citing appropriate literature.

REVIEWER COMMENTS

We thank the reviewers for their thorough evaluation of our manuscript and valuable feedback. Attached herewith is our point-by-point response addressing the comments provided by the referees.

Reviewer #3 (Remarks to the Author):

The reviewers have done a good job addressing my critiques and concerns. Good job!

We appreciate the reviewer's positive evaluation of our work. Thank you.

Reviewer #5 (Remarks to the Author):

Authors have satisfyingly addressed all my previous comments.

Thank you for recognizing our work.

Reviewer #7 (Remarks to the Author):

The manuscript entitled "Mitochondrial injury induced by a Salmonella genotoxin triggers the proinflammatory senescence-associated secretory phenotype" by Chen et al. has studied the effect of intoxication with the typhoid toxin on cellular senescence. The author has revised the paper according to the opinions of the reviewers, but there are still some minor problems that need to be corrected.

1. Figure 2h, Why is there no difference in cGAS between WT and PltbS35A after STING KO?

Thank you for the Reviewer's question. In typhoid toxin-treated macrophages, the upregulation of cGAS expression is likely due to the positive feedback regulation linked with type I IFN production. This feedback loop, as supported by the previous study (PMID: 25609843), indicates that type I IFN I positively modulates cGAS expression. Hence, the decrease in cGAS expression observed in STING KO cells following typhoid toxin treatment may explain the absence of noticeable.

2. The last three plots of Figure 6d are not labeled with detection index names.

We apologize for any confusion caused by the lack of detection index names in Figure 6d. We have rectified this by adding the index names to the figure.

3. Please provide verification of the effect of mitoQ on scavenging ROS in macrophages.

Thank you for the Reviewer's suggestion. We have included verification of the effect of mitoQ on scavenging ROS in THP-1-derived macrophages, shown in new Figure 5d. This result demonstrates that mitoQ treatment efficiently neutralized the overproduction of mitochondrial ROS in typhoid toxin-treated macrophages.

Reviewer #8 (Remarks to the Author):

A very interesting manuscript supported by high-quality data. Typhoid toxin penetration of mitochondria and the leakage of mitochondrial DNA into the cytosol leading to a SASP-like

response in macrophages is interesting and novel.

The authors have done a good job addressing reviewer comments. My only concern relates to point (iv) below. The authors state the macrophages are senescent but there is so much overlap with immunostimulated macrophages that it is difficult to determine whether the macrophages are senescent or undergoing some form of novel immunostimulation due to the toxin entering the mitochondria. The markers p16Ink4a/SA β G they use are at short timepoints (e.g. 16h) and are known to be reversed in macrophages after several days, which indicates they may in fact not be senescent, more senescent-like. This does not detract from the many interesting findings in the manuscript but I think the manuscript should be clear about this limitation in the discussion.

We appreciate the Reviewer's positive evaluation and insightful suggestion concerning the senescence state in macrophages. As per the Reviewer's recommendation, we have incorporated this aspect into the Discussion section of our manuscript. Thank you for your valuable input.

I have only minor suggestions to make the text more clear:

(i) line 30 in the abstract. 'essential genotoxin of Salmonella Typhi'. It is not clear what 'essential' refers to. Toxin-negative S.Typhi still cause typhoid fever in human participants (Gibani et al 2019) so the word 'essential' is misleading.

We appreciate the suggestion from the Reviewer. We have revised the sentence to ensure its accuracy aligns with the current evidence.

(ii) Following on from this, line 49-50 in the introduction: '...ethical considerations limit its in-depth investigation in human volunteer studies'. Human infection challenge studies have been reported (Gibani et al 2019). I think the reference should be included and/or sentence rephrased.

We thank the Reviewer's suggestion and apologize for this confusion caused. Prior studies utilizing a controlled human infection model have examined the role of typhoid toxin in disease pathogenesis. These studies typically involve volunteers being challenged with *Salmonella Typhi* and concluding the challenge period with antibiotics. While such investigations have advanced our understanding of the influence of typhoid toxin during the acute phase of infection, they may offer limited insights into its effects during chronic infection. To avoid any potential confusion, we have made revisions to the introduction section to provide clarity on this matter.

(iii) Line 116-119 - citation of Fig 1F relating to GSEA is missing

The citation related to Fig 1F has been promptly included in the article. Thank you for bringing it to our notice.

(iv) I tend to agree with reviewer 2, point 4 regarding the determination of senescence in the macrophages, which is still not clearly addressed experimentally and I suggest modest changes to the discussion to reflect this limitation. See below.

There is a lack of background or discussion on macrophage senescence, which appear similar to immunostimulated macrophages. Are the macrophage cells senescent? A defining feature of senescent cells is their durable cell cycle arrest in response to cell stress but this metric cannot be applied to the polarized THP1 macrophage. The authors observe

senescence markers p-p53, γ H2AX, and notably elevated p16Ink4a and SA β G. However, p16Ink4a/SA β G-signalling has been observed in macrophages as part of their natural physiological response to inflammatory stimuli and is reversed days after (<https://doi.org/10.18632/aging.101268>). Moreover, stimulated macrophages have an inflammatory secretome, which overlaps with the identified SASP factors, eg. IL6, IL1b, so it is perhaps not surprising a 'SASP-like' phenotype is enriched. There is significant debate regarding macrophage senescence and how it is defined (<https://doi.org/10.1083/jcb.202010162>). The manuscript contributes to this debate with their findings and this should be acknowledged and made clear within their discussion citing appropriate literature.

Thank you for the Reviewer's valuable input concerning the determination of senescence in macrophages, which contributes to enhancing the clarity of our study. Thus, we have made changes in the Discussion section to accurately reflect this limitation.